# Topology-Preserved Auto-regressive Mesh Generation in the Manner of Weaving Silk

**Gaochao Song**[1,2]    **Zibo Zhao**[3]    **Haohan Weng**[3]    **Jingbo Zeng**[1,2]
**Rongfei Jia**[4]    **Shenghua Gao**[1,2]*
[1]University of Hong Kong, [2]Shenzhen Loop Area Institute, [3]Tencent Hunyuan 3D, [4]Math Magic
https://gaochao-s.github.io/pages/MeshSilksong/

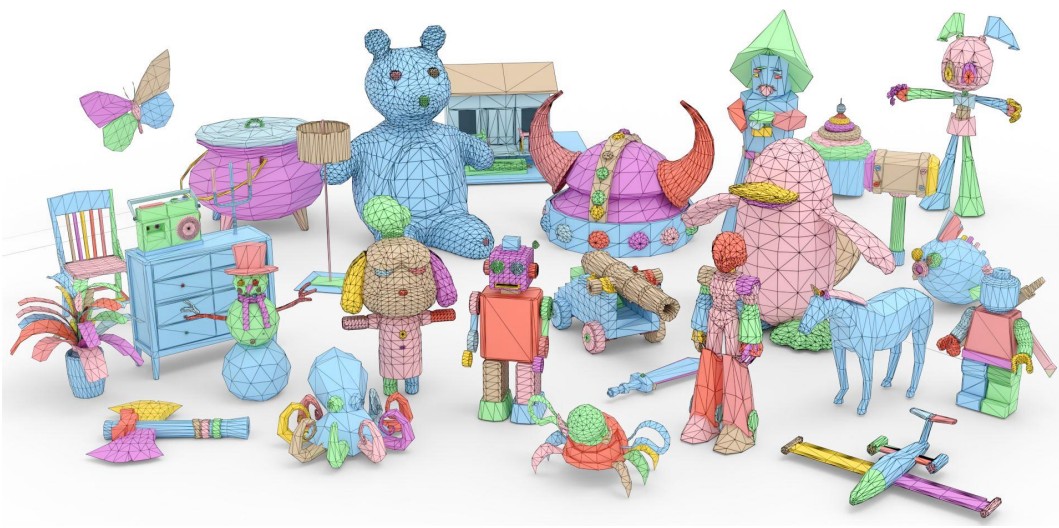

Figure 1: Meshes generated by our method. Vertices are colored based on different connected component of mesh. The generated meshes preserve geometric properties (manifoldness, watertightness, consistent face orientation, and part awareness) with state-of-the-art compression ratio.

## ABSTRACT

Existing auto-regressive mesh generation approaches suffer from ineffective topology preservation, which is crucial for practical applications. This limitation stems from previous mesh tokenization methods treating meshes as simple collections of equivalent triangles, lacking awareness of the overall topological structure during generation. To address this issue, we propose a novel mesh tokenization algorithm that provides a canonical topological framework through vertex layering and ordering, ensuring critical geometric properties including manifoldness, watertightness, face normal consistency, and part awareness in the generated meshes. Measured by Compression Ratio and Bits-per-face, we also achieved state-of-the-art compression efficiency. Furthermore, we introduce an online non-manifold data processing algorithm and a training resampling strategy to expand the scale of trainable dataset and avoid costly manual data curation. Experimental results demonstrate the effectiveness of our approach, showcasing not only intricate mesh generation but also significantly improved geometric integrity.

## 1 INTRODUCTION

The polygon mesh is a fundamental representation of 3D assets and is extensively utilized in video games, film production, and virtual reality. Generating high-quality meshes can significantly reduce

---

*Corresponding author.

Table 1: Summary of the advantages of our tokenization algorithm's geometric properties compared to baselines. Note that other tokenization methods including DeepMesh (Zhao et al., 2025a) and Nautilus (Wang et al., 2025) are based on BPT (Weng et al., 2025).

| Methods | Lossless | Manifold | Watertight | Face Normal Consistent | Part-aware |
|---|---|---|---|---|---|
| Ours | ✓ | ✓ | ✓ | ✓ | ✓ |
| MeshAnything v2 | ✓ | ✗ | ✗ | ✗ | ✗ |
| EdgeRunner | ✓ | ✓ | ✗ | ✗ | ✗ |
| TreeMeshGPT | ✗ | ✓ | ✗ | ✓ | ✗ |
| BPT | ✓ | ✗ | ✗ | ✗ | ✗ |

the workload of artists and improve industrial efficiency. Recent advancements (Siddiqui et al., 2024; Chen et al., 2024; Tang et al., 2025; Weng et al., 2025; Lionar et al., 2025; Zhao et al., 2025a) have focused on leveraging auto-regressive models to produce high-quality polygon meshes by directly learning vertex distributions and topological connectivity from handcrafted meshes. Such approaches (Weng et al., 2025; Zhao et al., 2025a) have yielded promising results.

However, the practical applications of auto-regressive mesh generation methods still face a critical challenge: the preservation of geometric properties. This limitation stems from previous mesh tokenization methods (Tang et al., 2025; Weng et al., 2025) treating meshes as simple collections of equivalent triangles and handle them equally during generation, which fail to strictly maintain application-friendly geometric properties, thereby restricting their practical utility. For instance, as shown in Tab.1, (1) these methods cannot guarantee the watertightness of generated meshes, as the algorithms lack real-time awareness of potential holes in the current triangle set. This property serves as a prerequisite for 3D printing, volume computation, and physics simulation. (2) They fail to ensure part awareness due to the absence of connected component concepts in algorithms, which is crucial for interactive editing, animation rigging, and downstream applications like robotics manipulation.

To address this issue, we propose a novel mesh tokenization algorithm akin to silk weaving. Inspired by geodesics on mesh, the algorithm's core lies in hierarchically layering vertices of each connected component in the mesh and sorting them layer by layer. This provides a canonical topological structure that reorganizes mesh vertices and connectivity to guarantee geometry properties during mesh generation. To achieve efficient compression, we design an adapted adjacency matrices compression algorithm that encodes each vertex using only 4 tokens. As shown in Tab.1, our algorithm is explicit, without any encoder-decoder architecture, thereby achieving lossless geometric compression while guaranteeing the manifold topology (Sec. 3.4 (1)). Furthermore, the algorithm inherently supports automatic watertightness detection during the mesh generation process (Sec. 3.4 (2)). It also rigorously ensures consistent normal orientation of adjacent faces (Sec. 3.4 (3)). Additionally, the algorithm supports part awareness, which enables the capture of small but important components during the auto-regressive generation (Fig.7). Finally, our algorithm achieves state-of-the-art performance in terms of both Compression Ratio and Bits-per-face metric (Tab. 2).

For data preparation, we propose an online non-manifold processing algorithm and a resampling training strategy. The former enables online processing after data augmentation (rotation and scaling), expanding training data and enhancing generalization, while existing methods like TreeMeshGPT (Lionar et al., 2025) cannot handle non-manifold data. The latter enables direct training on open-source datasets without manual curation while achieving comparable performance.

In summary, our contributions are as follows:

- We present a novel, efficient and lossless mesh tokenization algorithm achieving state-of-the-art compression efficiency (0.22 Compression Ratio and 26.65 Bits-per-face).

- Our algorithm inherently supports multiple geometric properties essential for downstream applications: manifold topology, watertightness detection, face normal consistency, and part awareness that are not simultaneously achieved by previous methods.

- For data processing, we propose a online non-manifold processing algorithm and a resampling training strategy. The former effectively expands the scale of trainable data and enhances model generalization, while the latter avoids time-consuming manual data filtering.

- Experimental results validate the effectiveness of our approach, highlighting its ability to capture small but critical connected components and improve overall geometric quality.

## 2 RELATED WORK

Previous works, including SDS-based (Li et al., 2024b; Lin et al., 2023; Tang et al., 2024; Raj et al., 2023; Yi et al., 2024), feed-forward-based (Hong et al., 2024; Li et al., 2024a; Xu et al., 2024), and VAEs + latent-diffusion-based (Chen et al., 2025c; Li et al., 2024c; Wang et al., 2023; Wu et al., 2024; Xiang et al., 2025; Yang et al., 2024; Zhao et al., 2025b; Zhang et al., 2023; 2024; Huang et al., 2025) methods typically represent 3D shapes using voxels or SDFs. Then they translate objects to dense triangle meshes using algorithms like Marching Cubes (Lorensen & Cline, 1998). This results in subpar geometric topology, characterized by poorly structured or cluttered wireframes.

To generate meshes with high-quality topology, recent methods leverage auto-regressive models to learn vertex distributions and connections directly from handcrafted meshes. MeshGPT (Siddiqui et al., 2024) pioneered this approach by encoding meshes into tokens using VQ-VAE (Van Den Oord et al., 2017), enabling direct supervision from handcrafted data. MeshXL (Chen et al., 2024) identified that quantizing triangle vertex tokens without trainable VQ-VAE is key to scaling datasets. Following this, (Chen et al., 2025a;b; Tang et al., 2025; Weng et al., 2025; Lionar et al., 2025; Hao et al., 2024) further optimized mesh tokenization algorithm or model architecture to support more faces. Other methods like LLaMAMesh (Wang et al., 2024) leveraged pretrained LLMs for text-to-mesh generation, while DeepMesh (Zhao et al., 2025a) applied reinforcement learning (Rafailov et al., 2023) to improve mesh aesthetics. However, as shown in Tab.1, due to flaws in the algorithm design, the practical applications of these methods still remain limited.

## 3 METHOD

Our algorithm consists of three sequential steps: (1) Vertex Layering and Sorting, which serves as the prerequisite for mesh-to-token compression; (2) Layer Adjacency Matrices Compression, which compresses vertex topological connectivity information into several tokens for optimal compression ratio; and (3) Token Packing, where vertex position tokens and topology tokens are organized into a unified format optimized for auto-regressive generation. As for data processing, we will detail our online non-manifold mesh processing algorithm and the resampling training strategy.

### 3.1 VERTEX LAYERING AND SORTING

The vertex layering and sorting algorithm is conducted on each connected component of the manifold mesh. We relabel the vertex as $\mathcal{V}_i^L$, where $L \in \{0, 1, 2, ...\}$ denotes the layer index and $i \in \{1, 2, ...\}$ denotes the order of vertex within that layer. As illustrated in Fig. 2-(a), given a connected component of the mesh, a starting half-edge $j\text{-}m$ can be uniquely determined by the $y$-$z$-$x$ coordinate ordering of all vertices, and we label $j$, $m$ as $\mathcal{V}_1^0$, $\mathcal{V}_1^1$ separately. Based on $\mathcal{V}_1^0$, the layer indices $L$ of all remaining vertices can be directly determined using Breadth-First Search (BFS) algorithm according to the shortest graph distance. Eventually, vertices at the same layer usually form a circle.

The vertex ordering follows a mathematical induction-like approach: the order of vertices in each layer is determined by the previous layer's ordering, starting from the first layer. Firstly, as shown in Fig. 2-(b), the vertices' order of the first layer can be directly obtained by traversing the neighboring vertices of $\mathcal{V}_1^0$ through the half-edge pointer: Since $\mathcal{V}_1^1$ is already determined by the starting half-edge, $\mathcal{V}_2^1$ is determined by the *previous twin* half-edge of $j\text{-}m$, and the remaining vertex order follows accordingly. Secondly, as illustrated in Fig. 2-(c), to determine vertices' order of the second layer, similar half-edge traversal is performed sequentially on each vertex in the first layer. For the vertices obtained through traversal, only those belonging to the second layer are retained. For example, among the neighbors of vertex $m$, only vertices $p$, $l$ are retained. Finally, as shown in Fig. 2-(d), all retained vertices are arranged together in queue order, and after deduplication, the complete ordering of all vertices in the second layer can be obtained. The same process is repeated for all remaining layers.

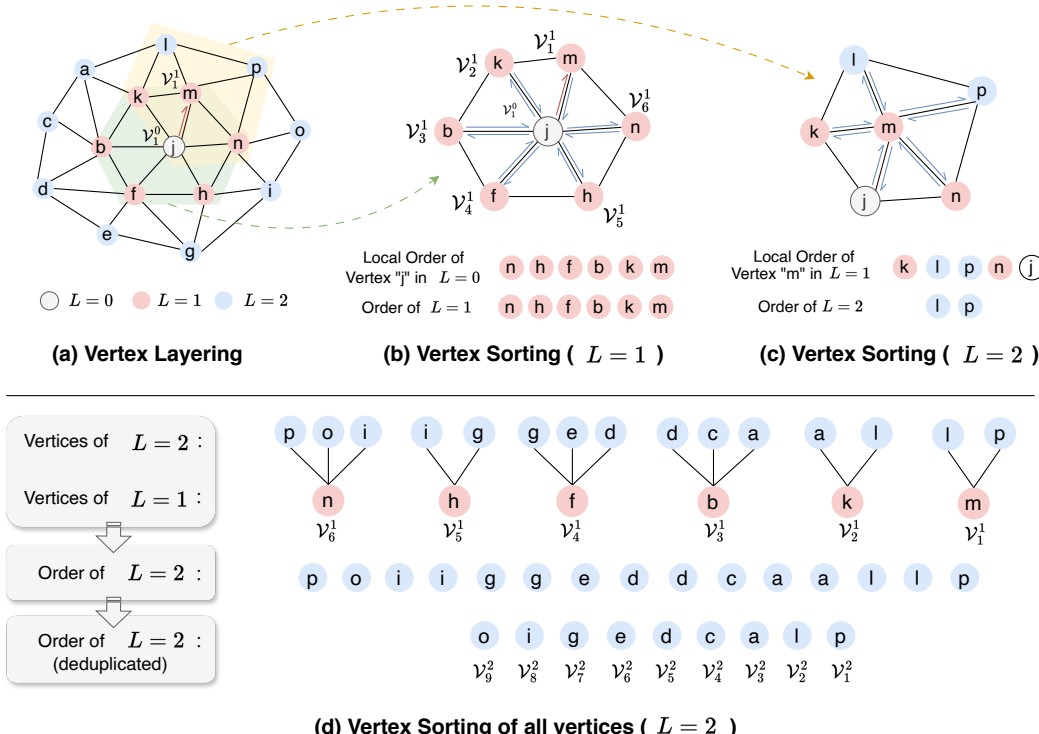

Figure 2: Illustration of vertex layering and vertex sorting. Please read the order from **right to left.** The vertex's layer can be obtained based on the shortest graph distance to starting point $j$, while the vertex ordering algorithm is performed in a manner similar to mathematical induction.

## 3.2 LAYER ADJACENCY MATRICES COMPRESSION AND TOKEN PACKING

After vertex layering and sorting, as illustrated in Fig.3, the connection of mesh vertices for each layer $L$ can be classified into two categories: self-layer connections (blue edges) and between-layer connections (red edges). The self-layer connections occur when both connected vertices belong to the same layer $L$, while between-layer connections link vertices between adjacent layers $L$ and $L-1$. Therefore, given vertices of layer $L$, we describe the related topology connections via two adjacency matrices named Self-Layer Matrix $\mathcal{S}_L$ and Between-Layer Matrix $\mathcal{B}_L$. In the next part, we will describe how to transform the matrices to tokens to compress the topology information.

**Self-Layer Matrix Compression.** Intuitively, self-layer connections can be directly determined through the ordered vertices without requiring additional tokens for representation. In statistics, such cases indeed account for more than 90%, but some other self-layer connections cannot be simply determined through vertex order. As shown in Fig.3, the edge $\mathcal{V}_2^L$-$\mathcal{V}_4^L$ belongs to such a special situation. Therefore, we uniformly represent all self-layer connections as Self-Layer Matrix and compress them together. The Self-Layer Matrix $\mathcal{S}_L$ is a 0-1 symmetric matrix with shape $M \times M$, where $M$ represents the vertex number for layer $L$. Directly using binary compression for each row is not feasible, as this would generate a vocabulary of size up to $2^m$, where $m$ represents the max number of vertices for all of the layers. Therefore, we set up a binary compression window with fixed size $W$. Considering that the Self-Layer matrix is symmetric, we slide the compression window one unit to the right for each row to avoid redundant compression. As illustrated in top-right of Fig.3, we first expand the symmetric matrix to $M \times 2M$, where the yellow block 0, 1 represent the regions to be compressed. The red window with size = 3 represents the binary compression window, which directly converts the matrix 0-1 values into decimal token indices. Note that even if there are regions not covered by the red window, such as the "1" at $(1, 6)$, its information can be recovered based on the content of $(6, 1)$ due to the matrix expansion. In statistics, we set $W = 8$, which can already cover 99.1% of self-layer connection cases. For the remaining extreme cases, we adopt a COO (Coordinate Format Representation)-like approach to ensure our algorithm can fully compress arbitrary 0-1 matrix, see Appendix B for more details.

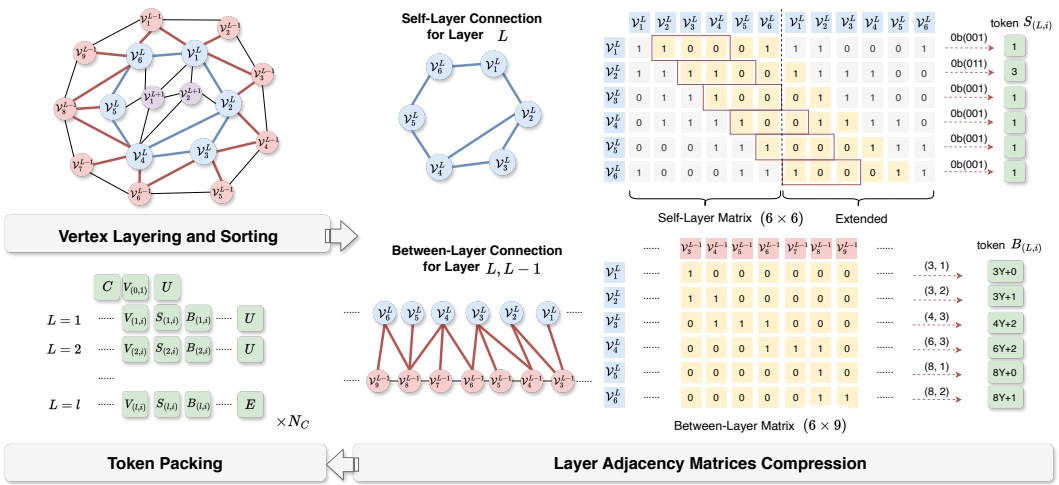

Figure 3: Illustration of our mesh tokenization algorithm. The mesh vertex for layer $L$ with order $i$ is denoted as $\mathcal{V}_i^L$, its corresponding three types of tokens are denoted as $V_{(L,i)}, S_{(L,i)}, B_{(L,i)}$, coming from vertex coordinate quantization, self-layer matrix $\mathcal{S}_L$'s compression and between-layer matrix $\mathcal{B}_L$'s compression respectively. The index $(i, j)$ for self-layer matrix $\mathcal{S}_L$ indicates the connection of $\mathcal{V}_i^L, \mathcal{V}_j^L$, while the index $(i, j)$ for between-layer matrix $\mathcal{B}_L$ indicates the connection of $\mathcal{V}_i^L, \mathcal{V}_j^{L-1}$.

**Between-Layer Matrix Compression.** The Between-Layer Matrix is a 0-1 matrix with size $M \times N$, where $M$ is the number of vertices for layer $L$ and $N$ is the number of vertices for layer $L-1$. Different from Self-Layer Matrix, we develop an algorithm similar to RLE (Run-Length Encoding) to compress the Between-Layer Matrix since we observed that "1"s tend to appear continuously in each row. Take bottom-right of Fig.3 as an example, there are continuous three "1"s at $(3, 4), (3, 5), (3, 6)$ for row 3, hence we firstly mark row 3 as $(x, y) = (4, 3)$, where $x \in [1, m]$ denotes the starting column index of consecutive ones, and $y \in [1, Y]$ represents the length of the consecutive ones sequence, with $Y$ being a predefined maximum length. Then the token index for $B_{(L,i)}$ should be $x \cdot Y + y - 1$. Up to this point, the RLE-like compression algorithm can already handle arbitrary 0-1 matrix. If multiple consecutive groups of "1"s appear in the same row separated by "0"s, the algorithm will simply use more than one token to represent the row information. To further reduce final token length, we upgrade the algorithm to solve an equivalent "Stars and Bars" question, aiming to compress each matrix row into only one token. See Appendix C for more details.

**Token Packing.** After performing matrices compression row by row, for each vertex $\mathcal{V}_i^L$, we now have three types of tokens: (1) vertex position tokens $V_{(L,i)}$; (2) self-layer topology tokens $S_{(L,i)}$; (3) between-layer topology tokens $B_{(L,i)}$. The $S_{(L,i)}, B_{(L,i)}$ are compressed from corresponding matrix row, typically contain one sub-token separately. The $V_{(L,i)}$ contains three sub-tokens, representing quantized $x$-$y$-$z$ coordinates. We further reduce these three sub-tokens to two by employing the block-offset representation in BPT (Weng et al., 2025). To obtain the full sequence for training, as illustrated in Fig.3, we start from special token $\boxed{C}$, which represents the start of connected component of mesh (If the mesh has $N_C$ connected components, there will be $N_C$ $\boxed{C}$ tokens). We pack $V_{(L,i)}$, $S_{(L,i)}, B_{(L,i)}$ together for vertex $\mathcal{V}_i^L$ and arrange different vertices' tokens following the layer order. Tokens for each layer are separated with a special "up-layer" token $\boxed{U}$ and the last one is replaced by another special token $\boxed{E}$ to represent the end of connected component.

### 3.3 Non-manifold Edges Processing and Training Strategy

**Non-manifold Edges Processing.** Non-manifold edges are defined as edges shared by three or more faces. Traditional non-manifold processing algorithms such as Libigl (Jacobson et al., 2013) first split all edges, then follow a degree-priority strategy, prioritizing the merging of edges shared by more faces; whereas our algorithm performs additional detection before edge merging to enhance surface integrity around non-manifold vertices. Fig.4 uses a minimal example to demonstrate the difference between our algorithm and Libigl. When performing edge merging, the Libigl algorithm may produce

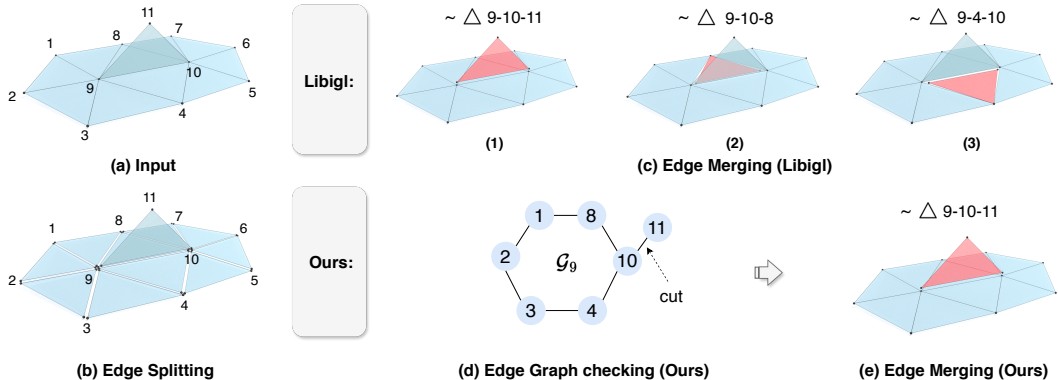

Figure 4: Difference between our non-manifold processing algorithm to Libigl (Jacobson et al., 2013). Our method performs additional edge structure checking around non-manifold vertex, ensuring surface integrity after edge merging. ~△ denotes the triangle that is ultimately separated.

three different results, as shown in (c), where (c)-1 is what we expect since the surface around vertices 9 and 10 remains intact, while (c)-2 and (c)-3 would compromise surface integrity, even though the non-manifold edges are also eliminated. Different from Libigl, we perform additional checking on non-manifold vertex 9 before edge merging, where the edges around it can be equivalently represented with an "edge graph" structure $\mathcal{G}_9$, as shown in (d). Note that the prerequisite for maintaining surface integrity around vertex 9 is that the edge graph forms a pure cycle, so we detach the edge 10-11 in edge graph, which is equivalent to detach △ 9-10-11 in mesh. It consequently produces only one edge merging result, as shown in (e). See Appendix A for more details.

**Training Strategy.** In the original open-source datasets, the face counts exhibit a significant long-tail distribution, meaning that simple meshes constitute the majority while complex meshes are in the minority. Without manual selection, the models tend to learn features from simple meshes, thereby affecting the ability to generate complex meshes. To avoid the high cost of manual selection, we draw inspiration from progressively-balanced sampling (Kang et al., 2020), classifying meshes based on face counts and adopting this resampling strategy during training. Specifically, we establish a class for every 100 face units, with index $j$. The sampling probability $p_j^{PB}(t)$ for class $j$ at epoch $t$ is:

$$p_j^{PB}(t) = (1 - t/T)\, p_j^{IB} + (t/T)p_j^{CB}$$

where $p_j^{IB}$ and $p_j^{CB}$ are the probabilities for instance-balanced and class-balanced sampling (Kang et al., 2020), $t$ is the current epoch, and $T$ is the total number of epochs. Early epochs focus on instance-balanced sampling ($t \to 0$), while later epochs prioritize class-balanced sampling ($t \to T$), enabling effective learning of long-tailed classes with higher face counts.

**Model Architecture.** The core model we use for training is a decoder-only transformer, each layer contains a cross-attention and a self-attention layer with a feed-forward network. For point cloud conditioned generation, the point cloud feature is extracted from Michelangelo (Zhao et al., 2023) and injected to auto-regressive model via cross-attention. The Michelangelo point cloud encoder and core model are trained jointly.

**Loss Function.** To train the auto-regressive model, we utilize the standard cross-entropy loss function, which minimizes the difference between the predicted token logits and the ground truth token sequence. The loss is expressed as:

$$\mathcal{L}_{\text{ce}} = -\sum_{t=1}^{T-1} S_{t+1} \log \hat{S}_t$$

where $\hat{S}_t$ refers to the predicted logits at time step $t$, and $S_{t+1}$ denotes the corresponding one-hot encoded ground truth token at the next time step.

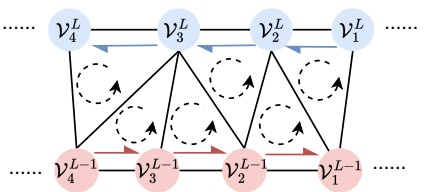

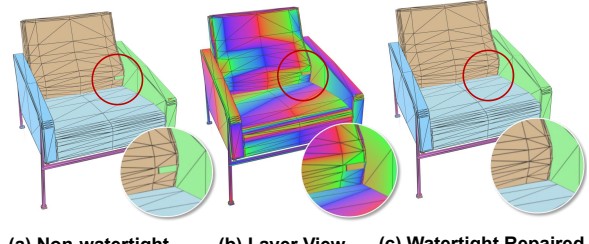

(a) Non-watertight    (b) Layer View    (c) Watertight Repaired

Figure 5: Illustration of face normal consistency. Half-edges of layer $L$ are pointing from lower order to higher order, while the layer $L-1$ is opposite.

Figure 6: Example of watertight detection and repair. The red-green-blue vertices in (b) denote the layer $3L$, $3L+1$, $3L+2$ respectively, where the missing faces are easily to be detected and repaired based on 0-1 matrix.

### 3.4 ANALYSIS OF GEOMETRIC PROPERTIES

**(1). Manifold Topology.** Our algorithm naturally enforces that edge connections during the generation process can only exist between vertices within the same layer or adjacent layers like weaving silk. During the decoding process, triangles are dynamically filled layer by layer in the same manner, thereby ensuring that the mesh strictly maintains a manifold topology throughout its layer-wise construction. The strict preservation of the manifold topology fundamentally stems from the constructive constraints applied during the mesh decoding process.

**(2). Watertight Detection and Repair.** While other methods may fail to detect erroneous tokens generated by auto-regressive models, potentially resulting in mesh holes, ours can well handle this. Take Fig.6-(b) as an example, we visualize each layer of vertices with red-green-blue contour lines. It is obvious that surface holes are invariably caused by certain entries in the self-layer matrix or between-layer matrix being incorrectly predicted as 0 instead of 1. Based on the connection rules in Sec. 3.2, such anomalies can be easily detected and the wrong tokens can be directly corrected.

**(3). Face Normal Consistency.** Our tokenization method ensures consistent face normal orientation without flipped triangles, as evidenced by the NC and |NC| metrics in Tab.2. As shown in Fig.5, triangles between layers always have at least two vertices in the same layer. By defining ascending half-edge directions for layer $L$ and descending directions for layer $L-1$, we guarantee counterclockwise vertex traversal for each triangle face. This approach ensures consistent normal orientations when computing face normals via cross-products of half-edge vectors.

## 4 EXPERIMENTS

### 4.1 COMPARISON SETTINGS AND RESULTS

**Datasets.** The model is trained on the mixture of gObjaverse (Zuo et al., 2024; Qiu et al., 2024) (a subset of Objaverse (Deitke et al., 2022) with ~280k meshes), ShapeNetV2 (Chang et al., 2015), 3D-FUTURE (Fu et al., 2021) and Toys4K (Stojanov et al., 2021) with around 100k meshes without manual selection. The meshes with token length > 10,000 and max vertex number per layer > 200 are filtered. Following TreeMeshGPT (Lionar et al., 2025), 1,000 meshes of gObjaverse and 200 meshes of other datasets are sampled and reserved for evaluation, with the remainder used for training.

**Metrics.** We apply four metrics to evaluate the point cloud conditioned generation quality, including Chamfer Distance (CD), Hausdorff Distance (HD), Normal Consistency (NC and |NC|). CD and HD are geometric metrics used to evaluate the similarity between point clouds or meshes. Normal Consistency (NC) assesses the alignment of surface normals, with higher values indicating better consistency. Its absolute form, |NC|, focuses purely on overall normal consistency.

**Implementation Details.** The decoder-only transformer has 24 layers and a hidden size of 1024, and the trainable parameter number is around 500M. It is trained on $16 \times$ H800 with around 15 days. The AdamW (Loshchilov & Hutter, 2017) is used as optimizer with $\beta_1$=0.9, $\beta_2$=0.99 and the learning rate starts from $1e^{-4}$ and decreases to $5e^{-5}$ based on cosine curve. More details including data augmentation, resampling settings are presented in the Appendix F.

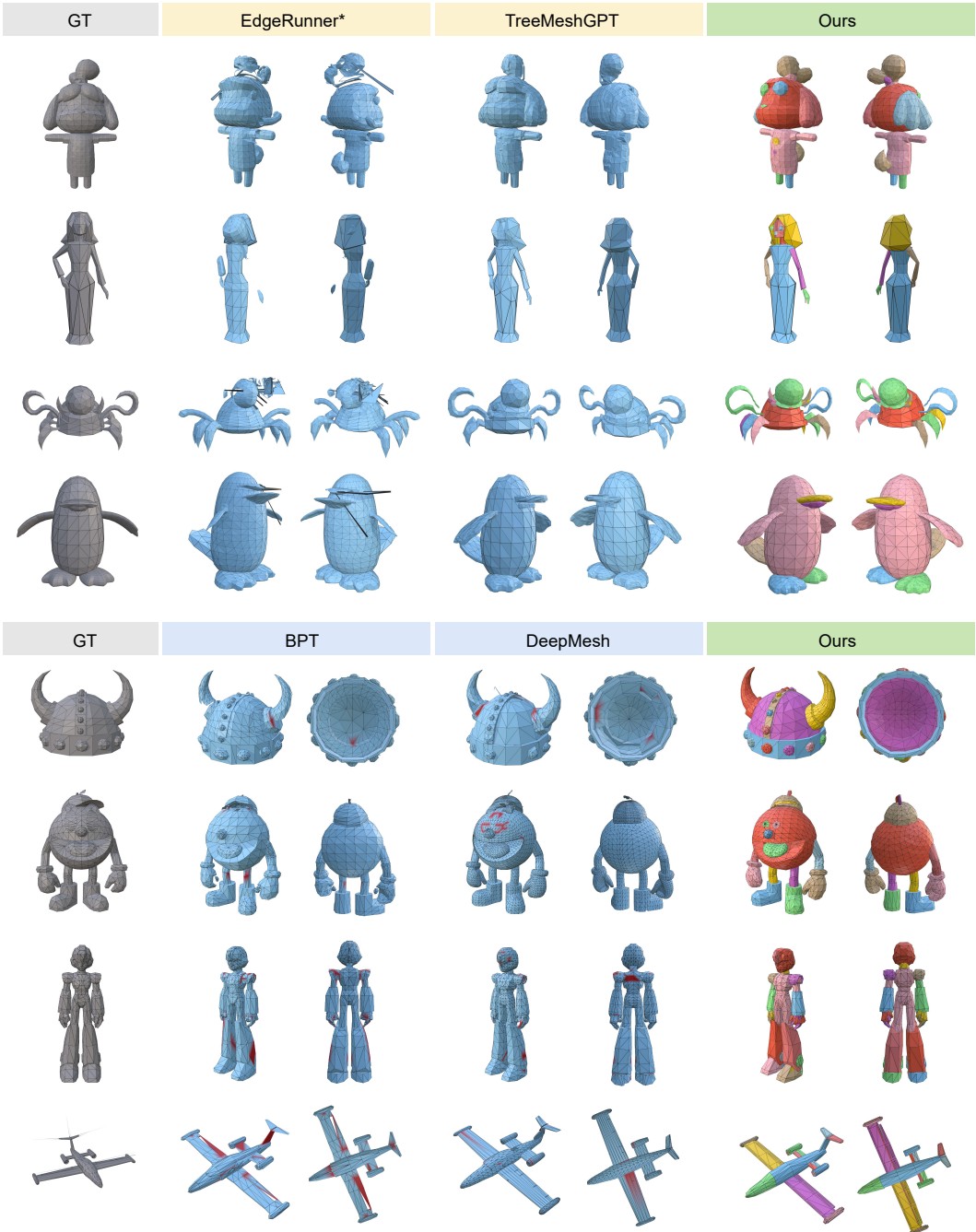

Figure 7: Qualitative Comparison on EdgeRunner* (Tang et al., 2025), TreeMeshGPT (Lionar et al., 2025), BPT (Weng et al., 2025), DeepMesh (Zhao et al., 2025a), and our method. The * denotes faithful training on the same dataset as ours. The connected components of our generated meshes are colored since our model is connected component aware based on token C and can capture details for minute connected components. (e.g., eyes of row 2, 6) (1). For tree-traversal methods EdgeRunner and TreeMeshGPT, the manifold topology for generated mesh is guaranteed, while our method demonstrates more robust generation capability and generates more details. (2). We achieved comparable visual results compared to local patch methods BPT and DeepMesh. However, these methods cannot generate meshes with manifold topology, as well as small components, which hinders practical application. The non-manifold edges are colored with red for BPT and DeepMesh.

Table 2: Quantitative comparison. The * denotes faithful training on the same dataset as ours. Our method offers higher geometric accuracy and normal consistency. Also, measured by Bits-per-face and Compression Ratio, we achieve SOTA compression efficiency. Refer to Appendix D for analysis.

| Method | CD ↓ | HD ↓ | NC ↑ | |NC| ↑ | Bits-per-face ↓ | Comp. Ratio ↓ |
|---|---|---|---|---|---|---|
| EdgeRunner* | 0.053 | 0.144 | 0.418 | 0.789 | 29.610 | 0.47 |
| TreeMeshGPT | 0.030 | 0.103 | 0.706 | 0.892 | 42.000 | 0.22 |
| BPT | 0.027 | 0.094 | 0.770 | 0.909 | 28.478 | 0.26 |
| Ours | **0.025** | **0.087** | **0.792** | **0.924** | **26.652** | **0.22** |

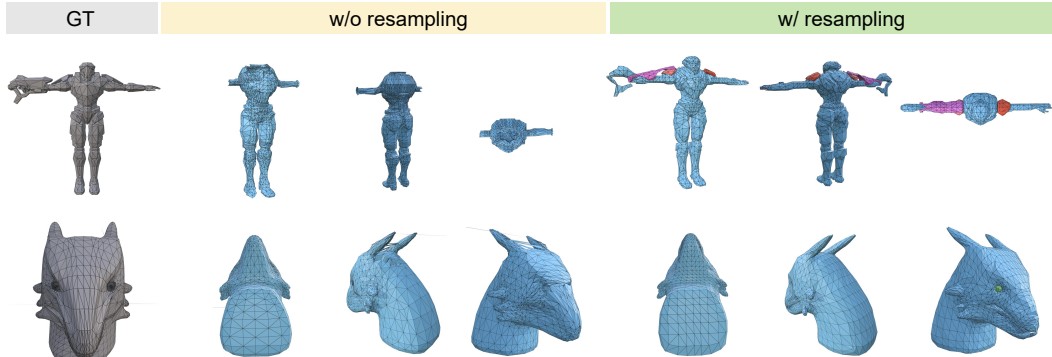

Figure 8: Ablation study on the resampling strategy. Our model achieves a more robust mesh generation capability on complex meshes with the resampling strategy than with the naive sampling.

**Qualitative Results.** We compare the qualitative results with EdgeRunner (Tang et al., 2025), TreeMeshGPT (Lionar et al., 2025), BPT (Weng et al., 2025) and DeepMesh (Zhao et al., 2025a). As shown in Fig. 7, our method shows robust generation capability, more mesh details, and strict manifold topology. We list our advantages in the figure caption.

**Quantitative Results.** To benchmark the capability of mesh generation, we use 500 meshes randomly sampled from reserved gObjaverse dataset for evaluation. We select EdgeRunner, TreeMeshGPT and BPT as the baselines. For EdgeRunner, we trained the model on the same dataset as ours and followed the released training setting with our resampling strategy. We also omitted the comparison with DeepMesh for fairness, as its tokenization algorithm is based on BPT and employs human feedback fine-tuning. The four quality metrics: CD, HD, NC, |NC|, and two compression metrics: Bits-per-face, Compression Ratio are presented in Tab. 2. Our method offers comparable geometric accuracy and better normal consistency, as well as SOTA compression efficiency.

## 4.2 ABLATION STUDY

**Resampling Strategy.** Our resampling plays an important role in generation quality since the training data displays a long-tailed distribution without manual selection. As illustrated in Tab. 3 and Fig. 8, the progressive resampling strategy enhances the model's ability to generate meshes with diverse and complex topological structures, avoiding overfitting to simpler object categories.

Table 3: Ablation study on the resampling strategy and non-manifold processing. The ablation study about non-manifold data processing is fine-tuned on a new Objaverse subset with ~50k items.

| Ablation | CD ↓ | HD ↓ | NC ↑ | |NC| ↑ |
|---|---|---|---|---|
| w/ resampling | **0.025** | **0.087** | **0.792** | **0.924** |
| w/o resampling | 0.032 | 0.103 | 0.700 | 0.880 |
| mani.+ non-mani. | **0.022** | **0.080** | **0.801** | **0.932** |
| mani. only | 0.027 | 0.090 | 0.688 | 0.871 |

**Non-manifold Processing.** Non-manifold mesh processing is instrumental for scaling datasets and improving generalization, as extensively validated in MeshXL and BPT. We compare the results of fine-tuning models on a new Objaverse subset using all data (~50k) versus using only manifold data (~13k), with 1000 samples used for evaluation. The Tab. 3 shows that using only manifold data severely limits the training data scale, resulting in a notable gap in generalization performance. Also, we present statistics of manifold and non-manifold data on four major datasets in Appendix G to illustrate the importance of non-manifold mesh processing in expanding training data scale.

**Window Size of Matrix Compression.** We also explore the influence of window size settings on matrix compression and vocabulary table, with relevant statistical data presented in Appendix G.

## 5 CONCLUSION

In summary, we propose a novel mesh tokenization algorithm with numerous application-friendly geometric properties and it also achieves the highest compression efficiency. We further introduce an online non-manifold processing algorithm to scale trainable data, along with a resampling training strategy that effectively avoids manual selection of high-quality data.

**Limitations.** Our algorithm requires a relatively high vocabulary size of up to 10,267, but under the bits-per-face metric, we still maintain the highest compression efficiency. Also, our method requires the predefinition of the maximum supported matrix size $m$, which corresponds to the maximum number of vertices per layer. It is designed to prevent long processing times for extreme meshes that could block online data iteration. We discussed this limitation specifically in Appendix H.

**Future Work.** Unlike other works that treat meshes as simple collections of triangles, we re-examine the mesh from a vertex-and-topology perspective, thereby achieving both excellent geometric properties and state-of-the-art compression ratio. However, there remains ample room for improvement in the current compression algorithm. Some promising directions include:

- Binary Matrix Factorization. Binary matrix factorization can be directly applied to the layer adjacency matrix, which involves assigning a binary vector to each vertex and then recovering the original layer adjacency matrix via Hamming distance or inner product. Compared to current matrix compression algorithms, this approach is mathematically more elegant and can theoretically achieve higher compression ratio (e.g., 2 tokens per vertex).

- Hybrid Polygon Support. The current algorithm inherently supports a hybrid representation of triangles and quadrilaterals, as it focuses solely on vertices and connectivity (the layer adjacency matrix) rather than triangles. Extending the algorithm to support hybrid polygon representations while maintaining robustness is a highly promising research direction.

## ACKNOWLEDGEMENTS

The work described in this paper was substantially sponsored by the General Research Fund (Project No. 17200725) supported by the Research Grants Council of Hong Kong and was partially supported by the Shenzhen Loop Area Institute. This work is also supported by Math Magic.

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

APPENDIX

INTRODUCTION OF APPENDIX

The appendix has several sections to introduce more details of our method, implementation details and experimental results.

- Appendix A introduces more complex scenarios of **non-manifold processing** in Sec. 3.3 of main paper body.
- Appendix B introduces some special case of **self-layer matrix compression** in Sec. 3.2 of main paper body.
- Appendix C introduces the full version of **between-layer matrix compression** in Sec. 3.2 of main paper body.
- Appendix D shows the analysis and derivation of our **compression ratio and Bits-per-face** in Tab.2 of main paper body.
- Appendix E shows more results of our **text-conditioned** mesh generation and **image-conditioned** mesh generation.
- Appendix F introduces more **implementation details** of our experiments.
- Appendix G introduces more **ablation studies** and related statistics.
- Appendix H shows discussion of our **limitation** and related statistics.
- Appendix I presents **pseudocode and complexity analysis** for our core algorithm in Sec. 3.1, 3.2 of main paper body.
- Appendix J shows the **qualitative comparison of our non-manifold processing** method to naive method.
- Appendix K shows the qualitative comparison of the **part generation ability** between our method and baseline.
- Appendix L shows the analysis of some **failure cases** of our current implementation.

## A  NON-MANIFOLD PROCESSING

In this section, we introduce our non-manifold processing algorithm for more complex scenarios. First, we present additional examples of non-manifold vertices along with their corresponding edge graph representations. To address these complex situations, we propose three key rules for the edge graph partitioning algorithm to ensure the integrity of the local surface: (1) Max cycle or max chain first. (2) Mesh connected component first. (3) Breadth-First-Search (BFS) processing. The rationale behind these rules is explained in details in the following subsections.

### A.1  EQUIVALENT TRANSFORM FROM MESH TO EDGE GRAPH

**Definition of Edge Graph.** We define the edge graph of a mesh vertex $i$ as a graph structure composed of all mesh edges connected to it, denoted as $\mathcal{G}_i$. In the edge graph, "graph nodes" represent vertices that form edges with $i$ in the mesh, and "graph edges" naturally represent triangles that have $i$ as a vertex in the mesh.

As illustrated in Fig.9, we present six different scenarios, encompassing both simple and complex examples. Scenario (a) represents the simplest case. In the equivalent edge graph, there are two connected components that can be directly partitioned. Scenario (b) is another common case, featuring three sub-cycles in the edge graph. Scenario (c) is a boundary case, as there are no cycles present. Scenarios (d)(e)(f) demonstrate more complex cases, involving several different cycles or chains.

### A.2  PARTITION RULE 1: MAX CYCLE OR MAX CHAIN FIRST

In our algorithm, scenario like (a) in Fig.9 is handled first to ensure that each edge graph contains only one connected component. Next, cycles or chains are detected, and the cycle with the maximum

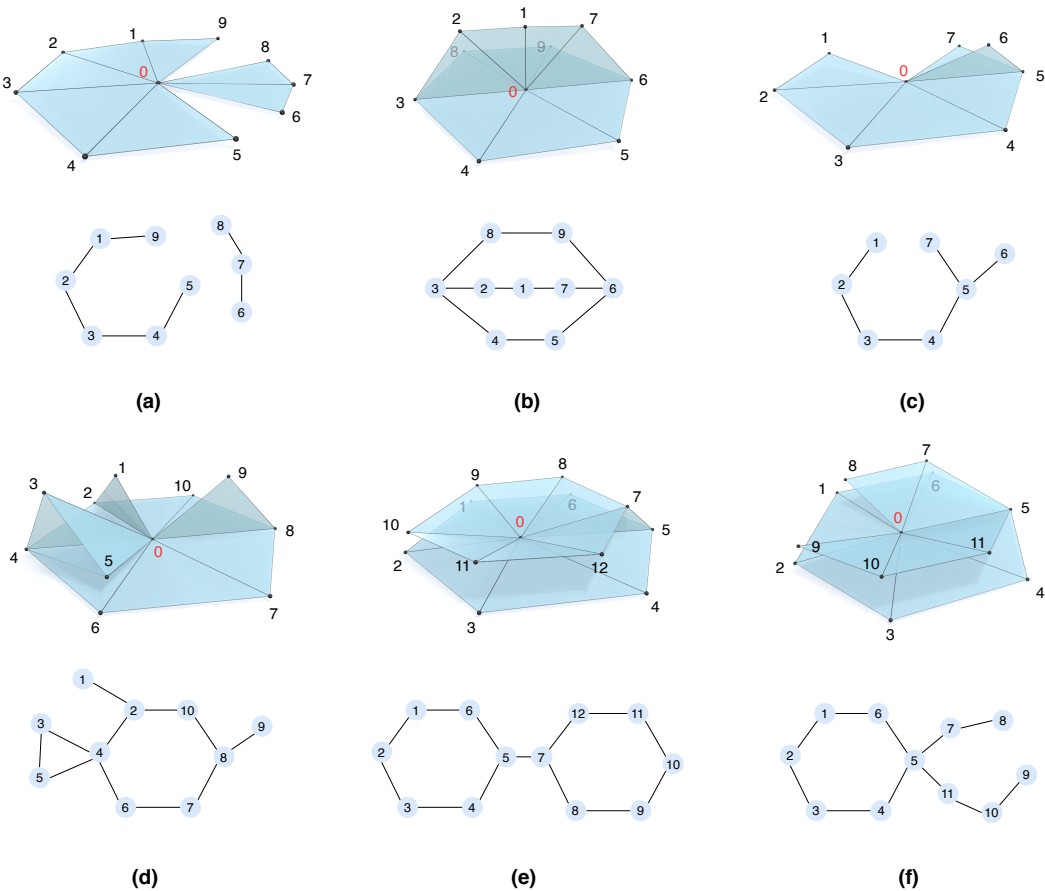

Figure 9: Illustration of six non-manifold scenarios. The non-manifold vertex 0 is marked as red, and all of each scenario is equal to an equivalent edge graph $\mathcal{G}_0$ under the mesh.

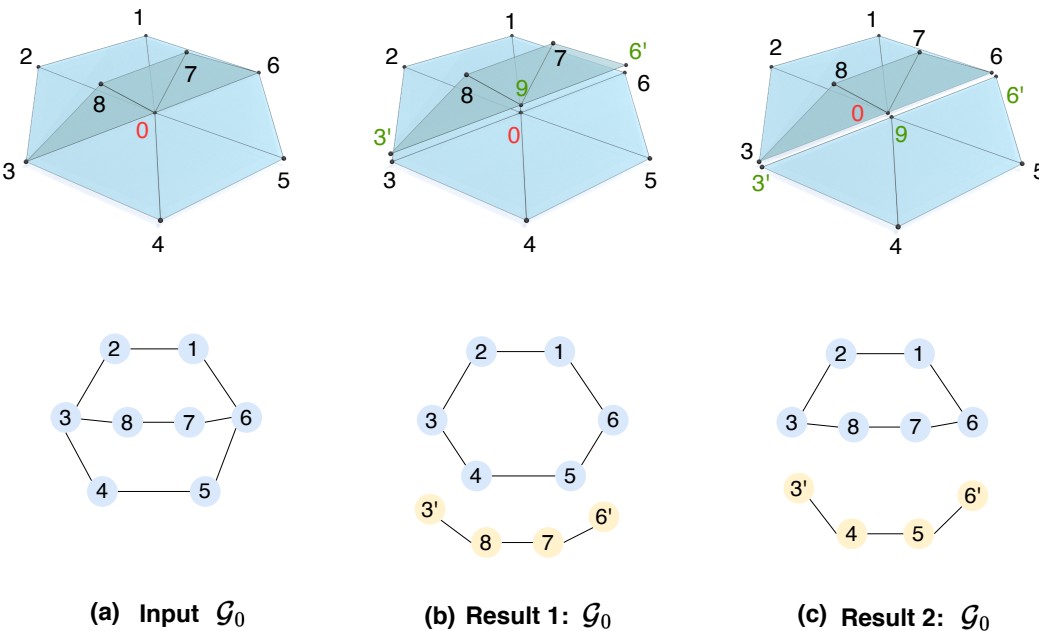

Figure 10: Illustration of Rule 1: Max Cycle or Max Chain First. Given input $\mathcal{G}_0$, we list two different partition results, and either of the result is reasonable temporarily. The vertex "9" indicates the new generated vertex due to edge graph partition, with same coordinate of vertex "0".

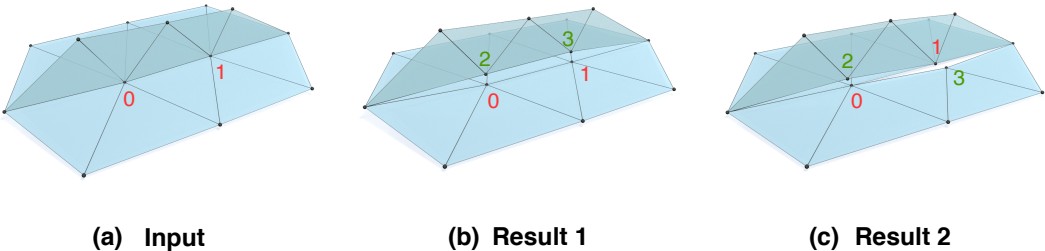

Figure 11: Illustration of the success case and failure case of Rule 1. The vertices "0","1" are processed based on Rule 1, with "2","3" are new generated vertices from edge graph partition. This will lead to two different results: (b) is success case since the completeness of surface is kept, while (c) is the failure case with broken surface.

length is selected and retained, while the remaining edges are detached. For example, in scenario like Fig.9 (d), the cycle 2-4-6-7-8-10-2 is retained, and the other edges will be detached. If no cycle exists, the chain with the maximum length is selected.

If there are two or more cycles of equal length in the edge graph, any of these cycles may be selected, and all such choices are considered reasonable. As illustrated in Fig.10, the input $\mathcal{G}_0$ contains three sub-cycles: 1-2-3-4-5-6-1, 1-2-3-8-7-6-1, and 3-4-5-6-7-8-3. We provide two different partition results, both of which are temporarily acceptable for our algorithm.

## A.3 PARTITION RULE 2: MESH CONNECTED COMPONENT FIRST

For a specific edge graph, Rule 1 may appear sufficient; however, when considering two connected non-manifold vertices, the completeness of the local surface is not guaranteed. As illustrated in Fig.11, applying the simple "max cycle or max chain first" rule can lead to two different partition results, as shown in (b) and (c). While (b) preserves surface integrity, (c) represents a failure case, as the surface integrity has been compromised. Therefore, the edge graph partitioning process should not only focus on a specific vertex but also consider its connected vertices.

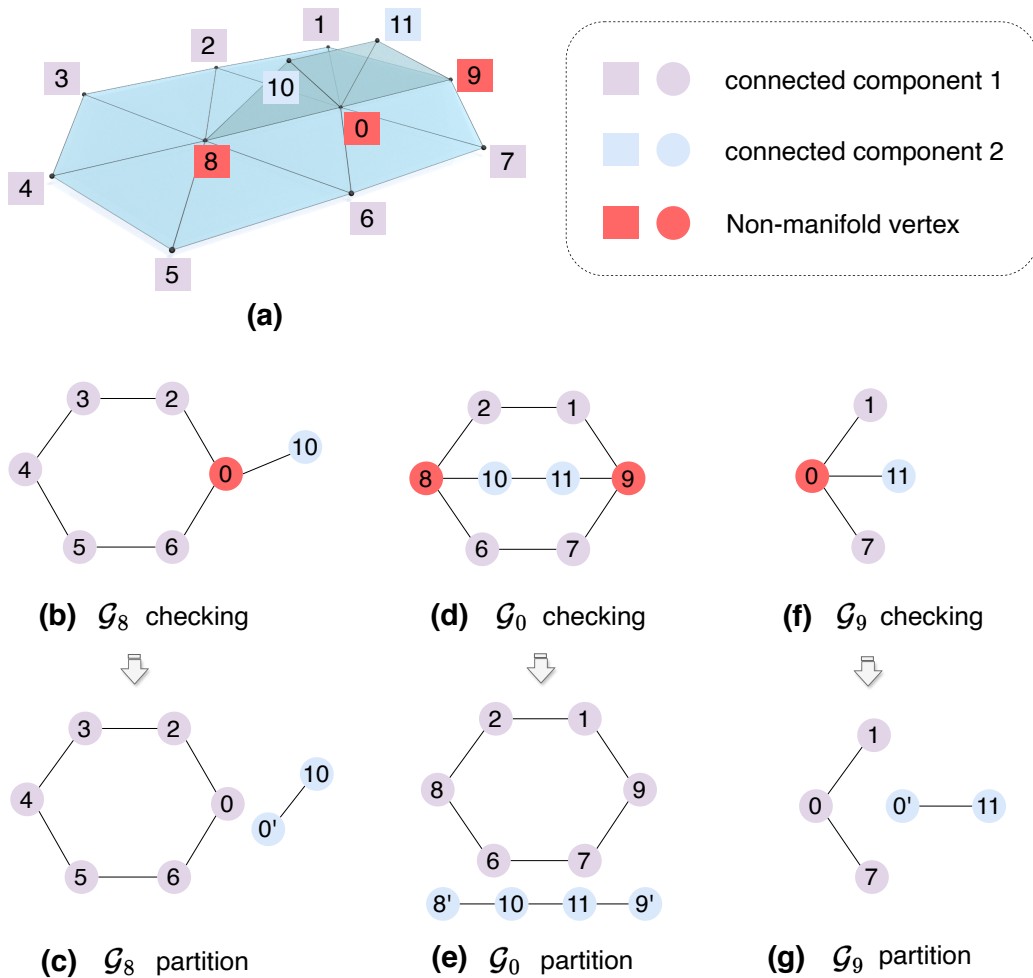

Figure 12: Illustration of Rule 2: Mesh Connected Component First. In the edge graph, the vertices belong to the same mesh connected component should be partitioned together as much as possible to keep the integrity of the whole mesh.

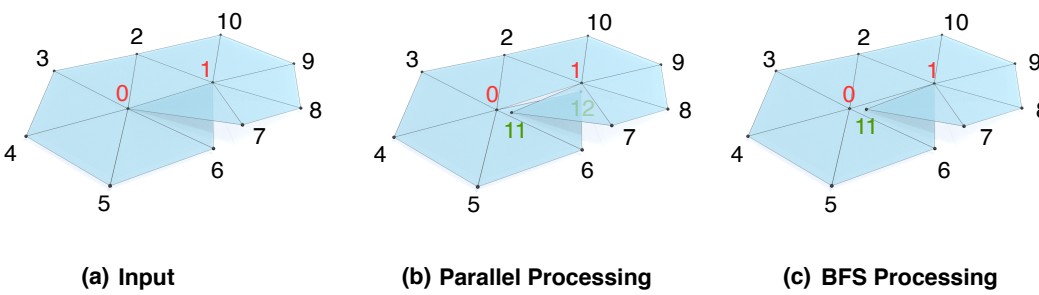

Figure 13: Illustration of parallel processing and Rule 3: BFS processing. Since the edge graph partition algorithm can affect surrounding vertices and potentially convert non-manifold vertices into manifold vertices, using the BFS algorithm to process them sequentially will result in less surface fragmentation.

To address this issue, we first assign a mesh connected component index (distinct from the connected components of the edge graph) to each manifold vertex, while marking non-manifold vertices with a special mesh connected component index. As illustrated in Fig.12 (a), all connected manifold vertices share the same mesh connected component index, where vertices {1,2,3,4,5,6,7} and {10,11} are grouped together, respectively.

Secondly, when checking the edge graph, vertices belonging to the same mesh connected component should be grouped together as much as possible, and this rule should take precedence over Rule 1. For example, in Fig.12 (d) and (e), when checking $\mathcal{G}_0$, three sub-cycles are available. However, the cycle 1-2-8-6-7-9-1 must be selected because vertices 1,2,6,7 belong to the same mesh connected component.

In the edge graph, non manifold vertices are marked as special mesh connected components (red color in Fig.12) and should ultimately be assigned to the mesh connected component of a manifold vertex. Therefore, the partition rule 2 should prioritize connecting edge graph vertices that belong to the same mesh connected component as much as possible.

### A.4 PARTITION RULE 3: BFS PROCESSING

After establishing Rule 1 and Rule 2, an unresolved issue remains: whether non-manifold vertices should be processed in parallel, specifically through parallel edge graph partitioning, given its higher computational efficiency. However, parallel processing may compromise surface integrity, as demonstrated in Fig.13. For instance, if non-manifold vertices "0" and "1" are processed in parallel, new vertices "11" and "12" will be generated, and the surfaces near "0" and "1" will become incomplete, as shown in (b).

To address this issue, we propose using the Breadth-First-Search (BFS) algorithm to sequentially process each non-manifold vertex. This sequential approach allows certain adjacent non-manifold vertices to be automatically corrected into manifold vertices, effectively minimizing the generation of redundant vertices and mitigating surface fragmentation. As shown in (c) of Fig.13, after processing vertex "0", vertex "1" is automatically corrected into a manifold vertex and does not require further processing, while the surface near vertex "0" maintains its integrity.

## B SELF-LAYER MATRIX COMPRESSION

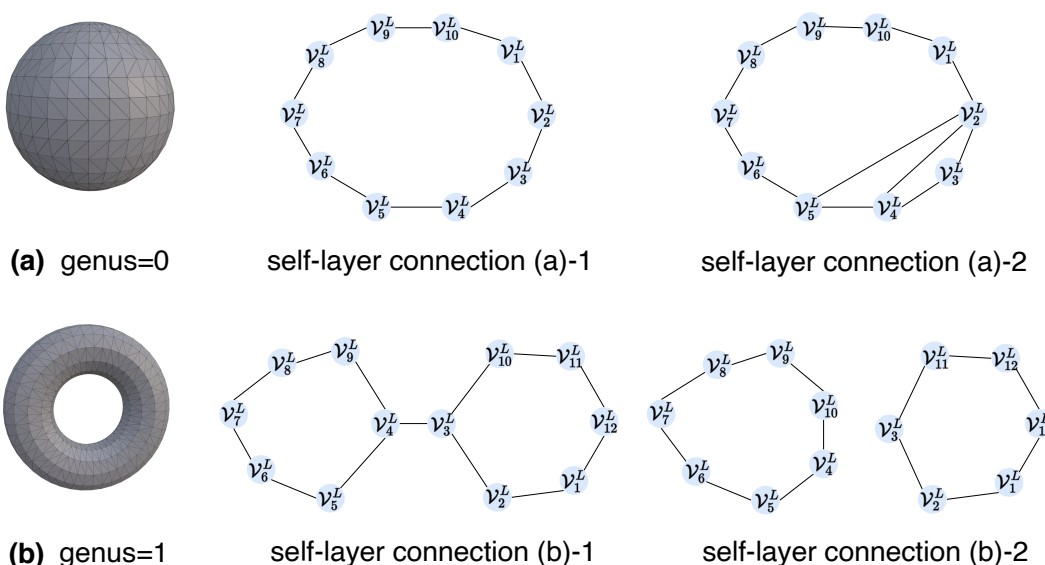

Figure 14: Illustration of several self-layer connections. When the genus=0, the self-layer connections are typically sequential, while the connections become complicated when genus>0.

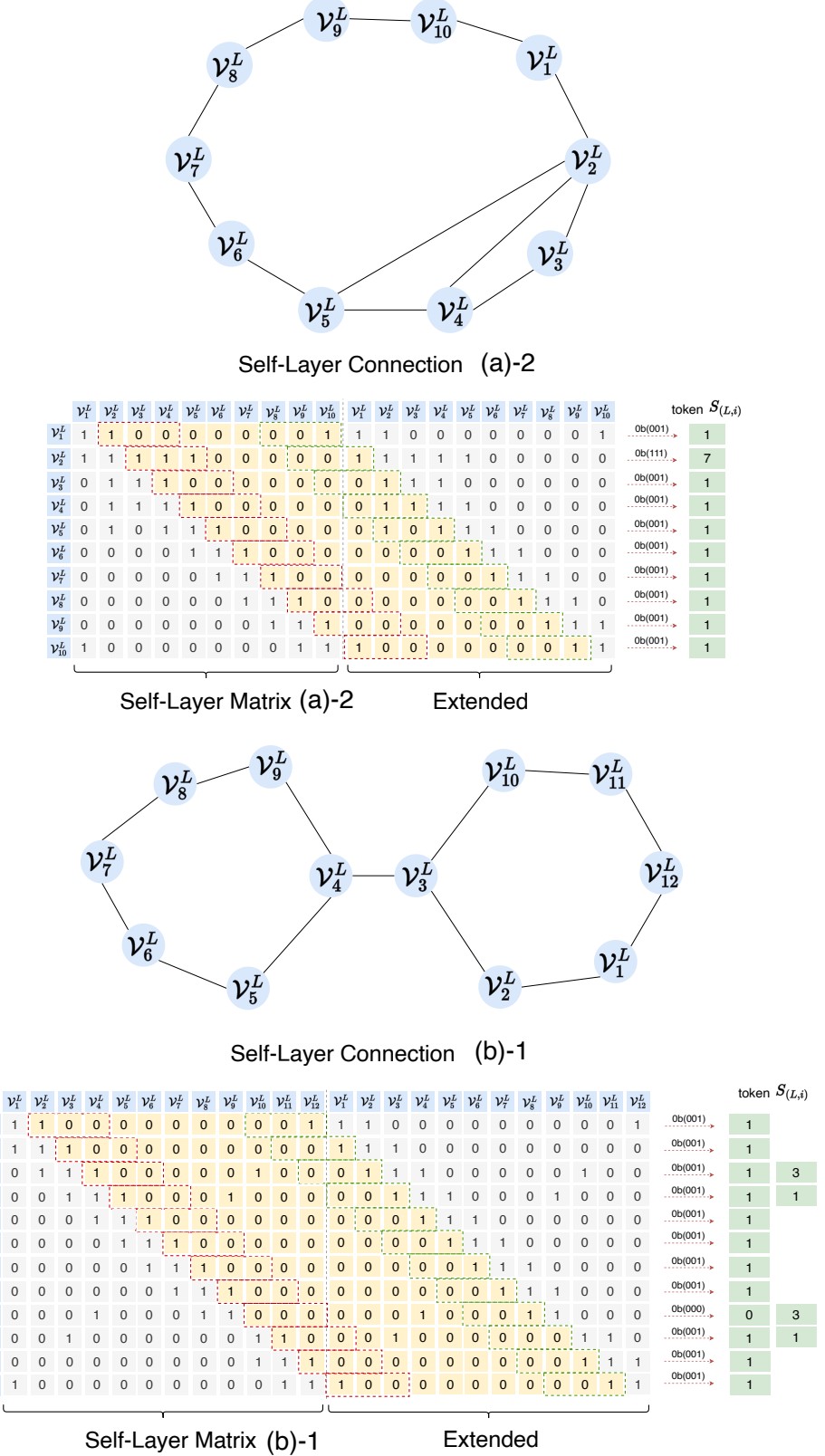

Figure 15: Illustration of self-layer matrix of (a)-2 and (b)-1 in Fig.14. The red box denotes compression window with size=3, while the green box denotes the redundant information and will be set to "0"s in practice. If "1"s occur outside red box and green box, the token number for the row will be > 1.

Generally, if a mesh is topologically homeomorphic to a sphere, the self-layer connections are typically sequential, as shown in Fig.14 (a)-1. In some occasional cases, several adjacent vertices may form local connections, as shown in Fig.14 (a)-2. When the topology of the mesh becomes more complex, such as having a genus of 1 (i.e., being homeomorphic to a torus), certain self-layer connections may exhibit configurations as illustrated in Fig.14 (b)-1, (b)-2.

To compress self-layer matrix, we take Fig.15 as an example. For sequential connections (a)-2, the matrix is easy to compress and the token number for each row is typically equal to one. In the self-layer matrix (a)-2, the red box denotes compression window with size=3, while the green box denotes redundant information and will be set to "0"s in practice. We notice the "1"s typically occur in the compression window since the connections are locally, and the token number for each row remains equal to one.

However, for the self-layer matrix (b)-1, the connections become more complex since several "1"s occur outside compression window. Since the compression window is fixed and does not slide, we use more tokens to compress the row. Take row 3 of self-layer matrix (b)-1 as an example, the "1" in position $(3, 10)$ lies outside the compression window and has an index distance of 3 (starts from 0), we use token index "3" to represent it.

Considering most of the rows still only have on token number, we use a vocabulary table with size $2^W$, where $W$ is the size of compression window. For rows with more than one token, we use an additional vocabulary table with size $m$ to represent extra tokens, where $m$ is the max number of layer vertices, following the same definition of main text.

In practice, we set $W = 8$ and $m = 200$, hence the totally vocabulary table size for self-layer matrix is $2^8 + 200 = 456$.

## C  BETWEEN-LAYER MATRIX COMPRESSION

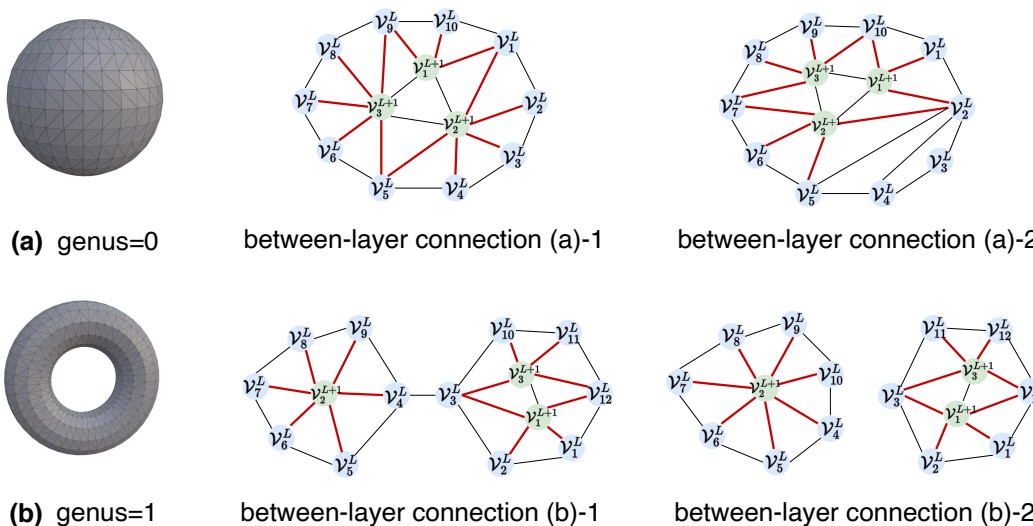

Figure 16: Illustration of between-layer connections for layer $L + 1$ and $L$. The red lines denote between-layer connections, which are related to self-layer connections of layer $L$.

Between-layer connections exhibit different patterns compared to self-layer connections, however, they are interrelated. We illustrate between-layer connections for layer $L + 1$ and $L$ with mesh genus =0 and =1 in Fig.16. The rules governing the between-layer connections for scenarios (a)-1 and (a)-2 are straightforward, while the rules in scenarios (b)-1 and (b)-2 are more complex.

To efficiently compress between-layer matrix of (a)-1 and (a)-2, we transform the problem into an equivalent "stars and bars" question. Firstly, as shown in Fig.17, we set a window size $W' = 5$, and place it at the first "1"s of each row, as highlighted by the red boxes. Notice the size of red box

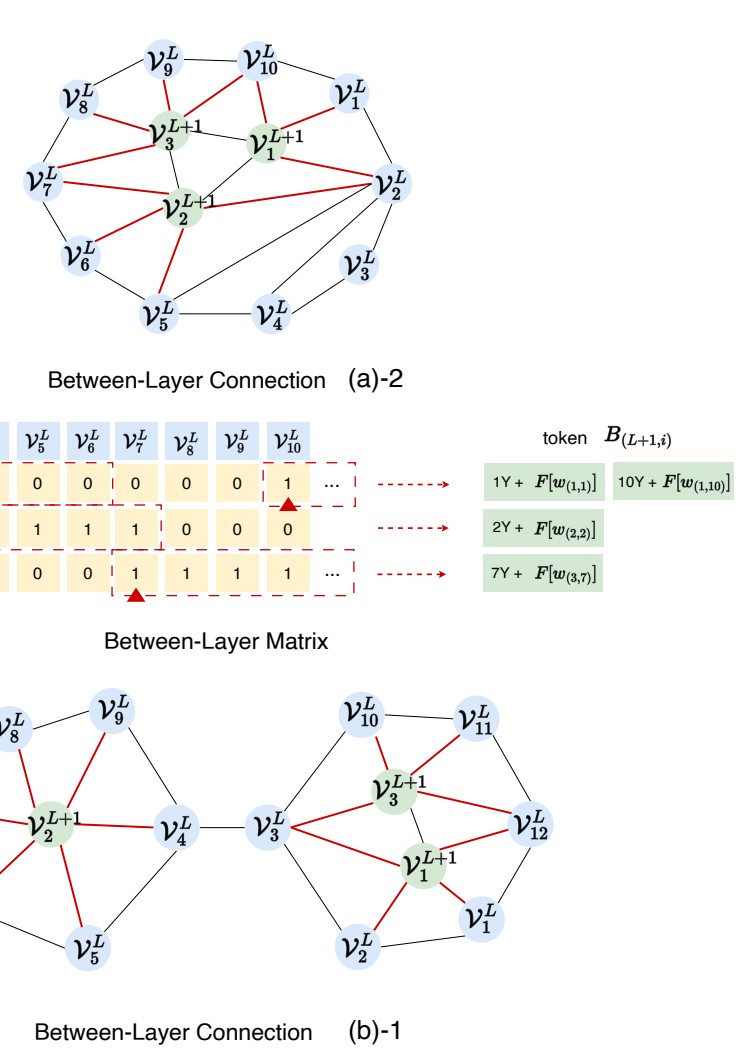

Figure 17: Illustration of between-layer matrix compression for layer $L+1$, $L$. $F[w_{(i,j)}]$ denotes the compressed token index of window (red box) located at $(i,j)$. We transform the compression of "0"s and "1"s inside the window into an equivalent "stars and bars" question, where Y denotes the totally combinatorial number.

| Scenario | Between layer window | Combinatorial number |
|:---:|:---:|:---:|
| "1" | 1 1 1 1 1 1 ★ ★ ★ ★ ★ | 1 |
| "0" | 1 0 0 0 0 0 ★ ★ ★ ★ ★ | 1 |
| "1"-"0" | 1 1 1 0 0 0 ★ ★ \| ★ ★ ★ | $C_{W'-1}^1$ |
| "1"-"0"-"1" | 1 1 0 0 1 1 ★ \| ★ ★ \| ★ ★ | $C_{W'-1}^2$ |
| "1"-"0"-"1"-"0" | 1 1 0 0 1 0 ★ \| ★ ★ \| ★ \| ★ | $C_{W'-1}^3$ |
| "0"-"1" | 1 0 0 0 1 1 ★ ★ ★ \| ★ ★ | $C_{W'-1}^1$ |
| "0"-"1"-"0" | 1 0 1 1 0 0 ★ \| ★ ★ \| ★ ★ | $C_{W'-1}^2$ |

Figure 18: Illustration of "stars and bars" question for between-layer matrix compression window. We list 7 scenarios, the window size $W'$ is 5.

is $W' + 1$ since the first element is always "1" and is excluded from consideration in subsequent calculations.

Secondly, after setting the compression window, we denote the finally compressed token index of the window at $(i, j)$ as $F[w_{(i,j)}]$, and transform the representation of "0"s and "1"s in the window to an equivalent "stars and bars" question. We notice in some cases, such as between-layer connections resembling (a)-2 in Fig.17, the pattern occasionally follows a "1"-"0"-"1" sequence due to the local self-layer connections in layer $L$. For example, this is evident in row 2 of the between-layer matrix. To address this, we classify the "stars and bars" question into seven distinct scenarios, as illustrated in Fig.18.

Finally, the total combinatorial number Y is:

$$Y = 2 \cdot W' + 2 \cdot C^2_{W'-1} + C^3_{W'-1}$$

and the total vocabulary table size of between-layer matrix is $m \cdot$ Y. In practice, we set the $W' = 5$ with Y=26, hence the finally vocabulary table size of between-layer matrix is 5200.

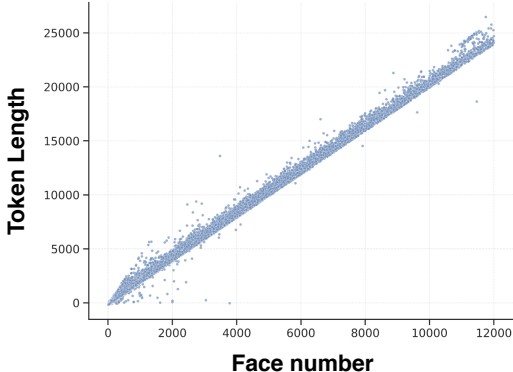

Figure 19: Statistic of the relationship between token length and the number of mesh faces on around 100k data samples. The ratio between them is approximately 2, indicating each face requires two tokens to be encoded on average.

## D    COMPRESSION RATIO AND BITS-PER-FACE ANALYSIS

We use two metrics to measure the compression efficiency of our algorithm: compression ratio and bits-per-face. For compression ratio, it is defined as the ratio of token length to 9 times the number of mesh faces, a metric that has been widely adopted by prior works (Tang et al., 2025; Weng et al., 2025; Zhao et al., 2025a; Lionar et al., 2025). For bits-per-face, it not only considers the total token length after compression but also accounts for potential compression ratio hacking through simply expanding vocabulary size, making it a more comprehensive metric for measuring compression efficiency. It is defined as:

$$\text{Bits-per-face} = \text{token-per-face} * \text{bits-per-token} = \text{token-per-face} * \log_2(\text{vocab size})$$

In this section, we will first introduce the statistics and derivation of compression ratio, then provide the specific calculation of Bits-per-face.

### D.1    STATISTIC OF COMPRESSION RATIO

As illustrated in Fig.19, we show the statistic result of token length and mesh face number of our dataset with around 100k items. The ratio between token length and face number is approximately equal to 2, meaning it requires 2 tokens to encode one face. Hence the compression ratio should be $2/9 \approx 0.22$.

### D.2    DERIVATION OF COMPRESSION RATIO

There is another way to derive the compression ratio $2/9$. Firstly, we suppose the human designed meshes are typically regular triangular meshes with degree equal to 6 on average (see D.3 for derivation). Secondly, we define 3 variables: (1) $N_f$: the face number of the mesh; (2) $N_v'$: the vertex number of mesh without deduplication; (3) $N_v$: the vertex number with deduplication.

It is obvious that the $N_v' = 3N_f$ since each triangle contains 3 vertices. Considering the average degree for a human designed regular mesh is 6, this implies each vertex is repeated 6 times. Therefore, we have the following approximate relationship: $N_v' \approx 6N_v$. Further, we have $N_v \approx 0.5N_f$.

Notice we utilize 4 tokens to compress a vertex, meaning the token length is around $4N_v$. Hence the compression ratio can be calculated as follows:

$$\frac{4N_v}{9N_f} \approx \frac{4 \cdot 0.5N_f}{9N_f} = 2/9$$

### D.3 DERIVATION OF AVERAGE DEGREE.

In a triangular mesh with many faces, the average vertex degree is closely related to the Euler characteristic (Tutte, 1963; Shewchuk, 2002; George & Borouchaki). The following derivation explains why the average degree approaches 6 in a high-quality triangular mesh with many faces.

#### 1. EULER'S FORMULA

For any planar graph or closed surface, Euler's formula is given by [1]:

$$V - E + F = \chi$$

where:

- $V$: Number of vertices,
- $E$: Number of edges,
- $F$: Number of faces,
- $\chi$: Euler characteristic ($\chi = 2$ for planar graphs).

In our method, the meshes are processed as manifold structure, and its topology can be regarded as a planar graph, with each edge is shared by only two faces.

#### 2. RELATIONSHIP BETWEEN FACES AND EDGES IN A TRIANGULAR MESH

In the manifold triangular mesh, each face has 3 edges, and each edge is shared by two faces. Therefore, the total number of edges $E$ is related to the number of faces $F$ as:

$$E = \frac{3F}{2}$$

#### 3. AVERAGE VERTEX DEGREE

The degree of a vertex is defined as the number of edges connected to it. The average vertex degree AvgDegree for the entire mesh is:

$$\text{AvgDegree} = \frac{2E}{V}$$

where $2E$ represents each edge has two vertices. In addition, the degree of a vertex can also be considered as the ratio of the number of vertices without deduplication ($2E$) to the number of vertices with deduplication ($V$). Substituting $E = \frac{3F}{2}$ into the equation, we get:

$$\text{AvgDegree} = \frac{2 \cdot \frac{3F}{2}}{V} = \frac{3F}{V}$$

#### 4. SUBSTITUTING EULER'S FORMULA

From Euler's formula, substituting $E = \frac{3F}{2}$, we can express $V$ in terms of $F$:

$$V - \frac{3F}{2} + F = 2$$

Simplifying:

$$V = \frac{F}{2} + 2$$

---

[1] https://en.wikipedia.org/wiki/Euler_characteristic

5. APPROACHING THE LIMIT

Substituting $V = \frac{F}{2} + 2$ into the equation for AvgDegree, we have:

$$\text{AvgDegree} = \frac{3F}{V} = \frac{3F}{\frac{F}{2} + 2}$$

As the number of faces $F$ becomes very large (i.e., as the mesh is refined), the term $\frac{F}{2} + 2 \approx \frac{F}{2}$. Therefore:

$$\text{AvgDegree} \approx 6$$

This result demonstrates in a well-designed triangular mesh, the average vertex degree approaches 6.

### D.4 BITS-PER-FACE METRIC

We provide a bits-per-face analysis for relevant methods with the following settings:

- Since we focus exclusively on lossless geometric compression, we replace VQ-VAE compression for vertices with naive $x,y,z$ representation to avoid potential VQ-VAE hacking.
- For consistency across comparisons, we standardize on 7-bit mesh quantization.

As shown in the Tab.4, despite having a larger vocabulary size (10,267) compared to baselines, our algorithm still maintain state-of-the-art compression efficiency under the more comprehensive Bits-per-face metric.

Table 4: Bits-per-face metric comparison of baselines.

| Method | token-per-face $* \log_2(\text{vocab size})$ | Bits-per-face $\downarrow$ |
|---|---|---|
| Ours | $2 * \log_2(10267)$ | **26.652** |
| BPT | $2.34 * \log_2(4608)$ | 28.478 |
| EdgeRunner | $4.23 * \log_2(128)$ | 29.610 |
| TreeMeshGPT | $6 * \log_2(128)$ | 42.000 |

## E   IMAGE CONDITIONED AND TEXT CONDITIONED GENERATION

For text-conditioned generation, we firstly utilize the Luma AI Genie [2] text-to-3D model to generate dense meshes from text prompts, then we sample around 4096 point clouds to conduct point cloud conditioned generation for our auto-regressive model. For image-conditioned generation, we utilize TRELLIS (Xiang et al., 2025) for image-to-3D generation, and sample 4096 point clouds for our model too. Fig.20 and Fig.21 demonstrate our high-quality and well-aligned results.

---

[2]https://lumalabs.ai/genie

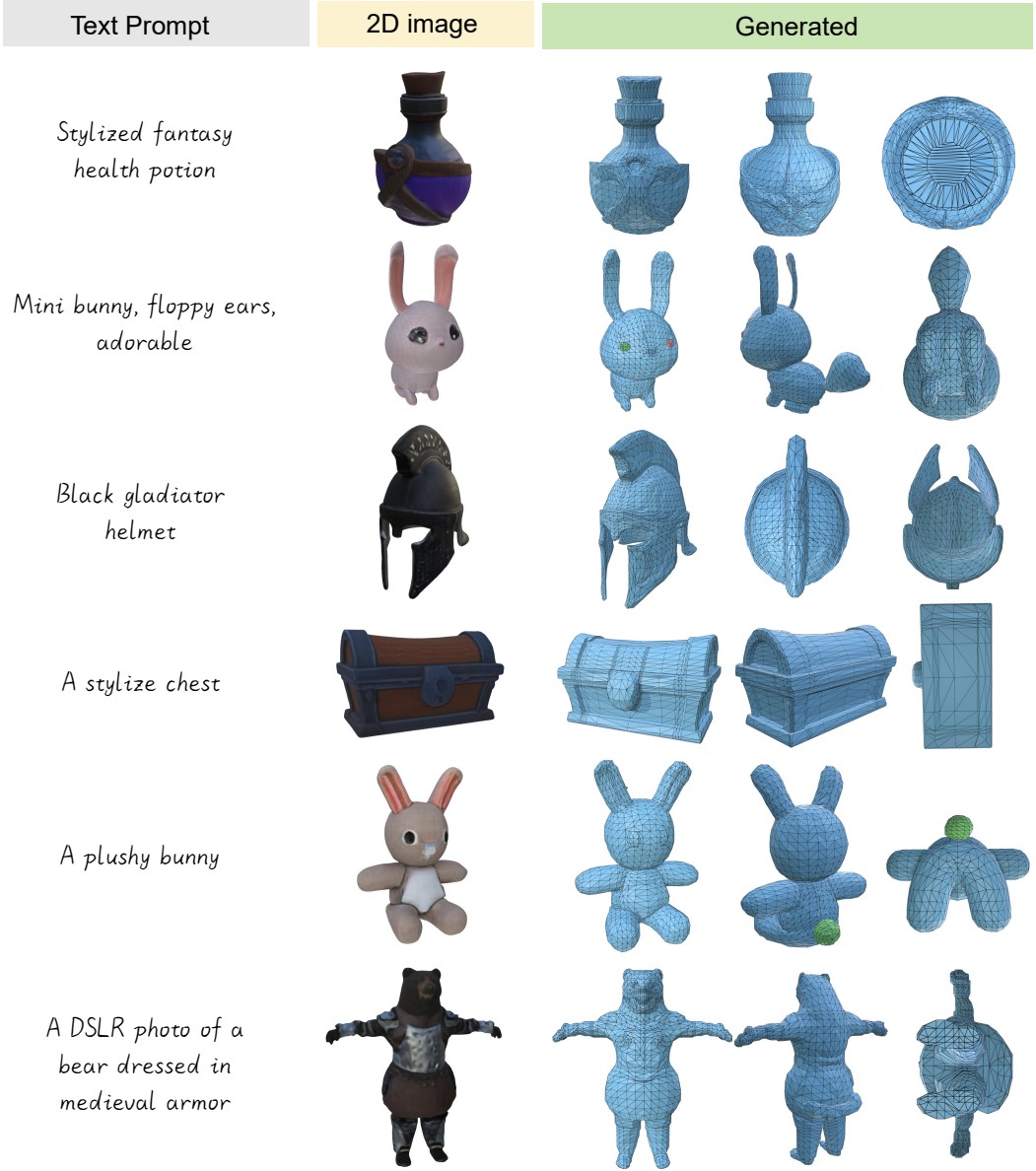

Figure 20: Illustration of our text conditioned generation. The Luma AI Genie text-to-3D model is utilized to generate dense meshes from text prompts, then the point clouds are sampled for point clouds conditioned generation.

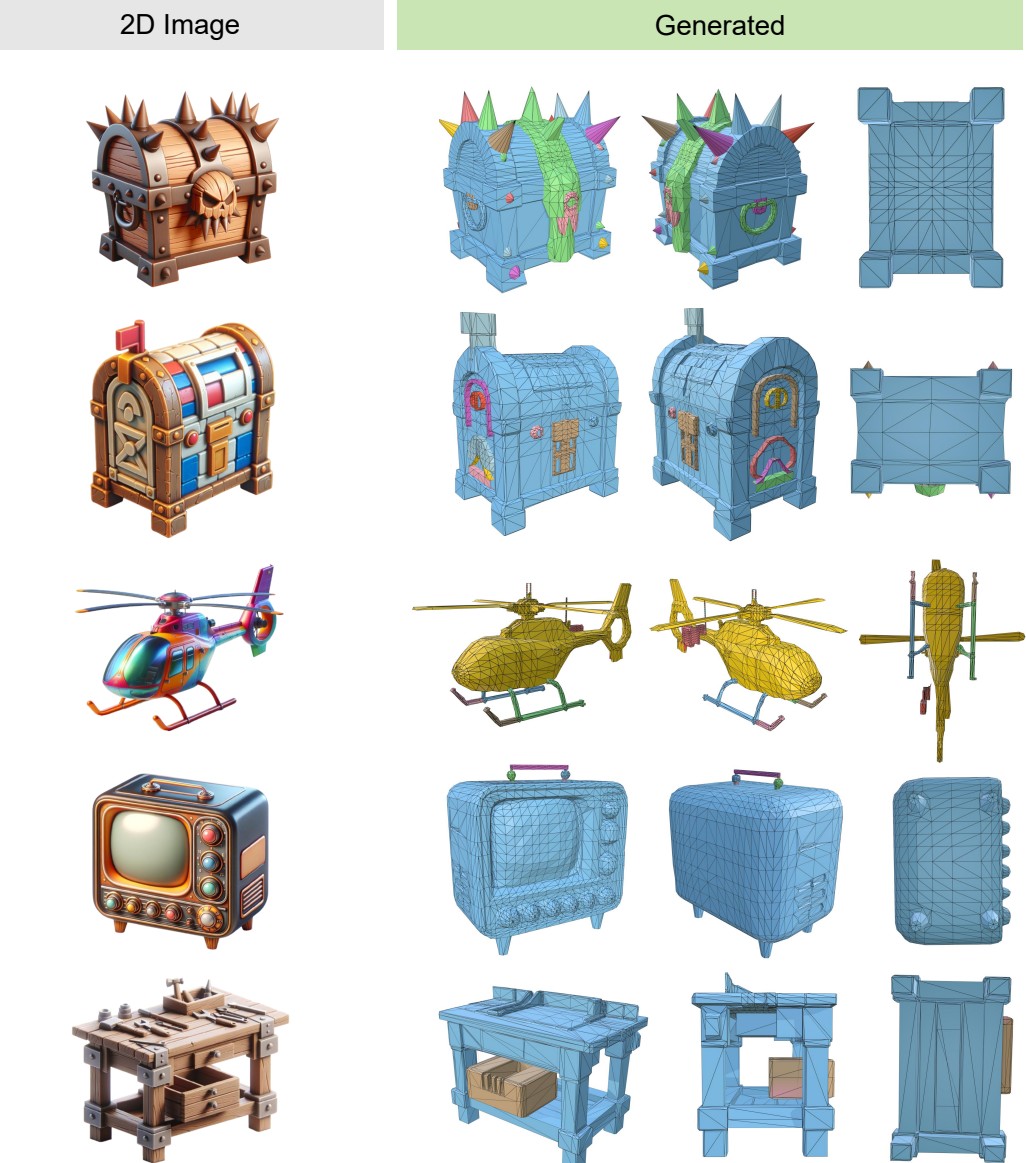

Figure 21: Illustration of our image conditioned generation. The TRELLIS (Xiang et al., 2025) is utilized for image-to-3D generation and point clouds of dense mesh are sampled for point clouds conditioned generation.

## F   MORE IMPLEMENTATION DETAILS

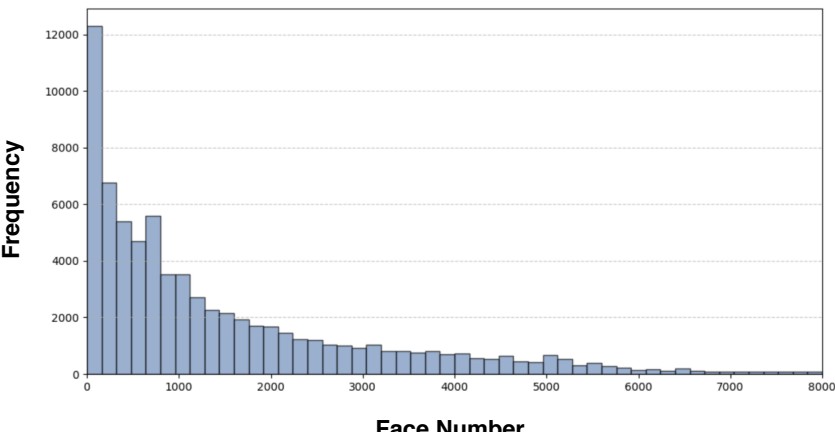

Figure 22: Illustration of the long-tail distribution of training data. The meshes with fewer faces constitute a larger proportion. Directly employing instance-balanced sampling may hinder the model's ability to effectively learn from meshes with a higher number of faces.

**Progressive-balanced Resampling.** As illustrated in Fig.22, our training data follows a long-tail distribution, hence we utilize a progressive-balanced resampling strategy during training, as described in Sec. 3.3. In early stages of training, we employ instance-balanced sampling (Kang et al., 2020) until the loss decreases to 0.37 with around 100 epochs. In the following 200 epochs, we control the transition from instance-balanced sampling to class-balanced sampling with training progress ratio from 0 to 1. The final converging loss is around 0.25.

**Data Augmentation.** To increase data diversity, we apply following augmentation: (1) Scaling: Each axis of mesh is scaled randomly with a factor chosen from the range $[0.75, 0.95]$. (2) Rotation: The mesh is set to perform rotational augmentation around the $y$-axis, with each unit being 30 degrees, covering a total of 360 degrees. For point clouds sampling, we sample 4096 points from each mesh as condition and added gaussian noise to enhance robustness with probability 0.5 during training.

Table 5: Statistics of non-manifold data on Objaverse, 3D-FUTURE, ShapeNetV2, Toys4K. As can be observed, non-manifold data constitutes a significantly high proportion across all datasets, highlighting the importance of non-manifold data processing for scaling up training data.

| Dataset | Manifold | Non-manifold | Manifold to Non-manifold |
|---|---|---|---|
| Objaverse (Deitke et al., 2022) | 0.261 | 0.564 | 0.175 |
| 3D-FUTURE (Fu et al., 2021) | 0.072 | 0.836 | 0.093 |
| ShapeNetV2 (Chang et al., 2015) | 0.143 | 0.787 | 0.069 |
| Toys4K (Stojanov et al., 2021) | 0.363 | 0.414 | 0.222 |

## G    MORE ABLATION STUDIES

**Non-manifold Data Statistics.** As shown in Tab.5, we present comprehensive statistics about the frequency of manifold/non-manifold meshes across four datasets under the assumption of 7-bit vertex quantization. The column definitions are:

- Manifold: meshes preserving manifold properties post 7-bit quantization.
- Non-manifold: inherently non-manifold meshes.
- Manifold to Non-manifold: originally manifold meshes that become non-manifold following 7-bit quantization-induced vertex merging.

The results demonstrate that across all datasets, non-manifold meshes constitute **over half of the data**. Furthermore, even originally manifold meshes exhibit a notable conversion rate to non-manifold topology as a consequence of vertex quantization.

**Window Size of Matrix Compression.** To assess how window size settings affect the degree to which each matrix row is compressed into a single token, we define a key metric "outside ratio" = "number of matrix rows with '1's falling outside the window" / "total number of matrix rows", and comprehensively analyze the impact of different window sizes on this metric using the Objaverse dataset (~250k items).

For Self-Layer Matrix, as shown in Tab.6, we have the following conclusion:

- With optimal window size selection ($W$=8), we achieve 99.1% coverage where all "1"s remain within the window boundaries.
- For out-of-window cases (0.9%), our algorithm still maintains lossless compression, the only cost is using more than one token to compress that row.
- Notably, $W = 1$ reduces to vertex connectivity determination based purely on sequential ordering, eliminating Self-Layer Matrix tokens while maintaining 90% coverage.

Table 6: The influence of $W$ in Self-Layer Matrix to "outside ratio" metric. * indicates our final parameter choice.

| $W$ | 1 | 2 | 3 | 4 | 5 | 6 | 7 | 8* | 9 | 10 |
|---|---|---|---|---|---|---|---|---|---|---|
| outside ratio | 0.100 | 0.055 | 0.039 | 0.031 | 0.025 | 0.020 | 0.0015 | 0.009 | 0.008 | 0.006 |
| vocab size | 1 | 4 | 8 | 16 | 32 | 64 | 128 | 256 | 512 | 1024 |

For Between-Layer Matrix, as shown in Tab.7, we have the following conclusion:

- Optimal window size selection ($W' = 5$) ensures 98.1% in-window coverage for "1"s.
- Out-of-window scenarios are still handled through lossless compression with using more than one token for that row.
- For $W' = 2, 3$, the "Stars and Bars question" simplifies to consecutive "1"s counting, achieving only 71.7% coverage.

Table 7: The influence of $W'$ in Between-Layer Matrix to "outside ratio" metric. * indicates our final parameter choice.

| $W'$ | 2 | 3 | 4 | 5* | 6 | 7 |
|---|---|---|---|---|---|---|
| outside ratio | 0.622 | 0.283 | 0.043 | 0.019 | 0.017 | 0.014 |
| combination number | 2 | 3 | 15 | 26 | 42 | 64 |
| vocab size | 400 | 600 | 3000 | 5200 | 8400 | 12800 |

## H  FURTHER DISCUSSION OF LIMITATION

Table 8: The frequency of maximum vertex number per layer ($m$) for each mesh on ~250k data items of Objaverse (0-16k faces). By setting $m = 200$, we can cover ~99.3% data items.

| Interval | [0, 40] | [40, 80] | [80, 120] | [120, 160] | [160, 200] | [200, ...] |
|---|---|---|---|---|---|---|
| Interval frequency | 0.603 | 0.240 | 0.085 | 0.049 | 0.016 | 0.007 |
| Cumulative frequency | 0.603 | 0.843 | 0.928 | 0.977 | 0.993 | 1.000 |

Table 9: The frequency of the maximum layer number for each mesh on ~250k data items of Objaverse (0-16k faces). By setting limitation = 200, we can cover ~99.95% data items.

| Interval | [0, 40] | [40, 80] | [80, 120] | [120, 160] | [160, 200] | [200, ...] |
|---|---|---|---|---|---|---|
| Interval frequency | 0.8925 | 0.0933 | 0.0120 | 0.0013 | 0.0004 | 0.0050 |
| Cumulative frequency | 0.8925 | 0.9858 | 0.9978 | 0.9991 | 0.9995 | 1.0000 |

Our designed matrix compression algorithm can process sparse 0-1 matrices of **arbitrary** sizes and configurations, with necessary optimizations implemented to support online processing tasks such as data augmentation. However, since the number of vertices per layer varies across different meshes, some extreme meshes with excessive vertex counts in a single layer would generate oversized matrices, causing blockages in online data iteration. Therefore, we define a parameter $m$, representing **the maximum allowed number of vertices per mesh layer**, which also corresponds to the maximum allowable matrix size for processing. Meshes exceeding this threshold are discarded during iteration. As shown in Tab.8, we set a reasonable threshold of $m = 200$, covering up to 99.3% of the data while ensuring unblocked data iteration. In practice, we have comfortable average processing time of approximately 0.3s-0.8s per mesh, allowing iteration approximately every 1 second.

We also provide the statistic of frequency of **the maximum layer number** for each mesh in Tab.9. We also set the layer number limitation = 200 in practice, which can cover ~99.95% data items.

# I PSEUDOCODE AND ALGORITHM COMPLEXITY ANALYSIS

## I.1 PSEUDOCODE

We present the pseudocode for our core tokenization algorithm below, encompassing: start half-edge initialization, BFS-based vertex layering and sorting, and matrix compression. Please note that in the actual algorithm implementation, vertex layering and sorting are performed simultaneously for acceleration.

---

**Algorithm 1:** Our Mesh Tokenization Algorithm.

---

**Data:** Mesh $M$ with connected components $\{CC_1, CC_2, \ldots, CC_k\}$
**Result:** vertex tokens and topology tokens for each connected component
**foreach** *connected component $CC_i$ in $M$* **do**

    // 1. Get start half-edge
    $v_0, v_1 \leftarrow$ sort_half_edges$(CC_i)[0]$ // represent start half-edge as
        $v_0 \rightarrow v_1$
    // 2. BFS Vertex Layering and Sorting
    $L \leftarrow 0$ // layer number
    $v_0$.layer $\leftarrow 0$, $v_1$.layer $\leftarrow 1$;
    $Q[L] \leftarrow \{v_0\}$ // $Q[L]$ is the sorted vertices sequence for layer $L$
    **while** *True* **do**

        $Q_{\text{last}} = Q[L]$;
        $L + +$;
        **foreach** *v in $Q_{last}$* **do**
            $v$.layer $\leftarrow L$;
        **end**
        **foreach** *v in $Q_{last}$* **do**
            $Q[L]$.append$(\{\ldots\}) \leftarrow \{\ldots\} \leftarrow$ get_sorted_next_layer_vertices$(v)$
            $Q[L]$.remove_duplicate_vertices();
        **end**
        **if** $Q[L] = \emptyset$ **then**
            break;
        **end**
        **if** $len(Q[L]) > 200$ **then**
            abandon this mesh;
        **end**
    **end**
    // 3. Matrix Compression
    $S_{\text{token}}[0] = \emptyset$ // $S_{\text{token}}[L]$ is self-layer matrix tokens for layer $L$
    $B_{\text{token}}[0] = \emptyset$ // $B_{\text{token}}[L]$ is between-layer matrix tokens for layer
        $L$
    **for** *layer* $= 1, 2, \ldots, L$ **do**
        $Q_{\text{last}} = Q[\text{layer} - 1]$;
        $Q_{\text{current}} = Q[\text{layer}]$;
        $S_{\text{token}}[\text{layer}] =$ compress_self_layer_matrix$(Q_{\text{current}})$;
        $B_{\text{token}}[\text{layer}] =$ compress_between_layer_matrix$(Q_{\text{current}}, Q_{\text{last}})$;
    **end**
**end**

---

## I.2 TIME AND SPACE COMPLEXITY ANALYSIS

Given a mesh, we assume:

- $K$: The number of connected component.
- $V$: The vertex number for each connected component.
- $F$: The face number for each connected component.
- $E$: The edge number for each connected component.

- $L$: The layer number for each connected component.
- $D$: The average degree for each vertex.
- $Lv$: The vertex number for each layer, i.e. $Lv * L = V$
- $m(m = 200)$: Maximum number of vertices per layer, i.e. $Lv < m$.

### I.2.1 VERTEX LAYERING AND SORTING

(1). It requires $O(E \log(E))$ for sorting the half-edges for each connected component.

(2). For each vertex with degree $D$, it requires $O(D)$ to get neighbor vertices' order that belong to next layer. The order is obtained from "previous" and "twin" pointer of half-edge data structure.

(3). It requires $O(L * Lv * D)$ for vertex layering and sorting. Note that $L * Lv = V$, and $D = 6$ for manifold mesh (See Appendix D.2 for derivation), hence $O(L * Lv * D) = O(V)$.

Based on the analysis:

- The time complexity is $O(K(E \log(E) + V))$.
- The space complexity is $O(K(V + E))$ since we use half-edge data structure.

### I.2.2 SELF-LAYER MATRIX COMPRESSION

To compress each row, the 0-1 numbers in the window is directly transformed from binary to decimal with $O(1)$, and we simply traverse each row and column for compression.

Based on the analysis:

- Time complexity is $O(K * L * Lv^2)$.
- Space complexity is $O(K * L * 2 * Lv^2) = O(K * L * Lv^2)$ since we concatenate a repeated matrix along the column for the convenience of compression. However, this does not affect the space complexity.

### I.2.3 BETWEEN-LAYER MATRIX COMPRESSION

To compress the 0-1 number in window, we predefined a static mapping table to enable $O(1)$ calculation. We simply traverse each row and column for compression.

Based on the analysis:

- Time complexity is $O(K * L * Lv * Lv') = O(K * L * Lv^2)$, where $Lv'$ represents vertex number of last layer.
- Space complexity is $O(K * L * Lv * Lv' + \text{size(mapping table)}) = O(K * L * Lv^2)$. The "size(mapping table)" is a fixed combinatorial number for "stars and bars" question and this will not affect the space complexity.

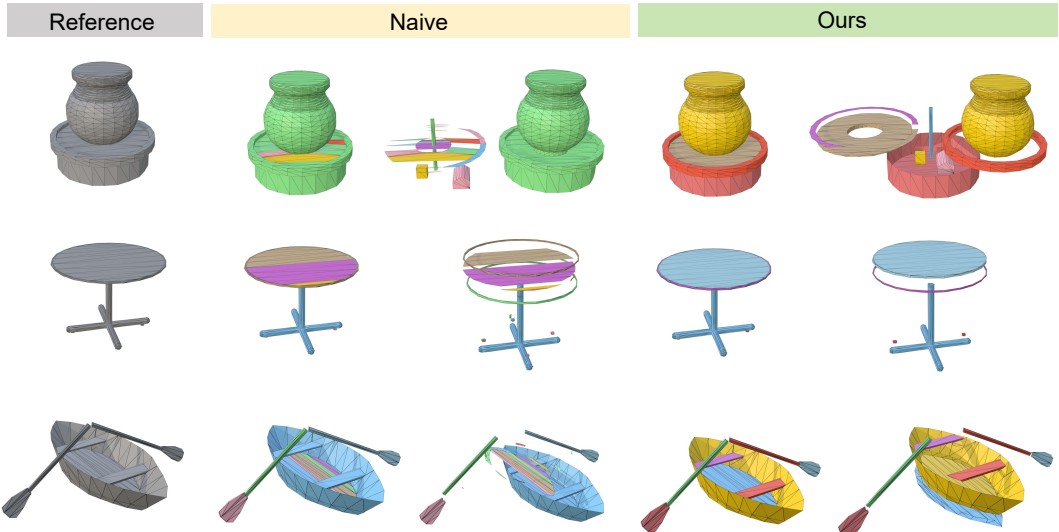

Figure 23: Comparison of non-manifold processing results on ShapeNetV2 (Chang et al., 2015). Different colors denote distinct connected components (left), which are spatially separated for clarity (right). Our method produces fewer fragmented triangles and more complete object components than the naive method.

## J QUALITATIVE COMPARISON OF NON-MANIFOLD PROCESSING

Our non-manifold processing algorithm performs additional "edge graph" detection during edge merging operations to maximally preserve surface integrity around non-manifold vertices. Overall, we effectively reduce the generation of fragmented triangles and obtain more complete manifold meshes.

As illustrated in Fig.23, we visualize the processed non-manifold mesh with different colors distinguishing different connected components. The left side shows the overall processing result, while the right side shows the result after offsetting the connected components to facilitate the differentiation of distinct object components.

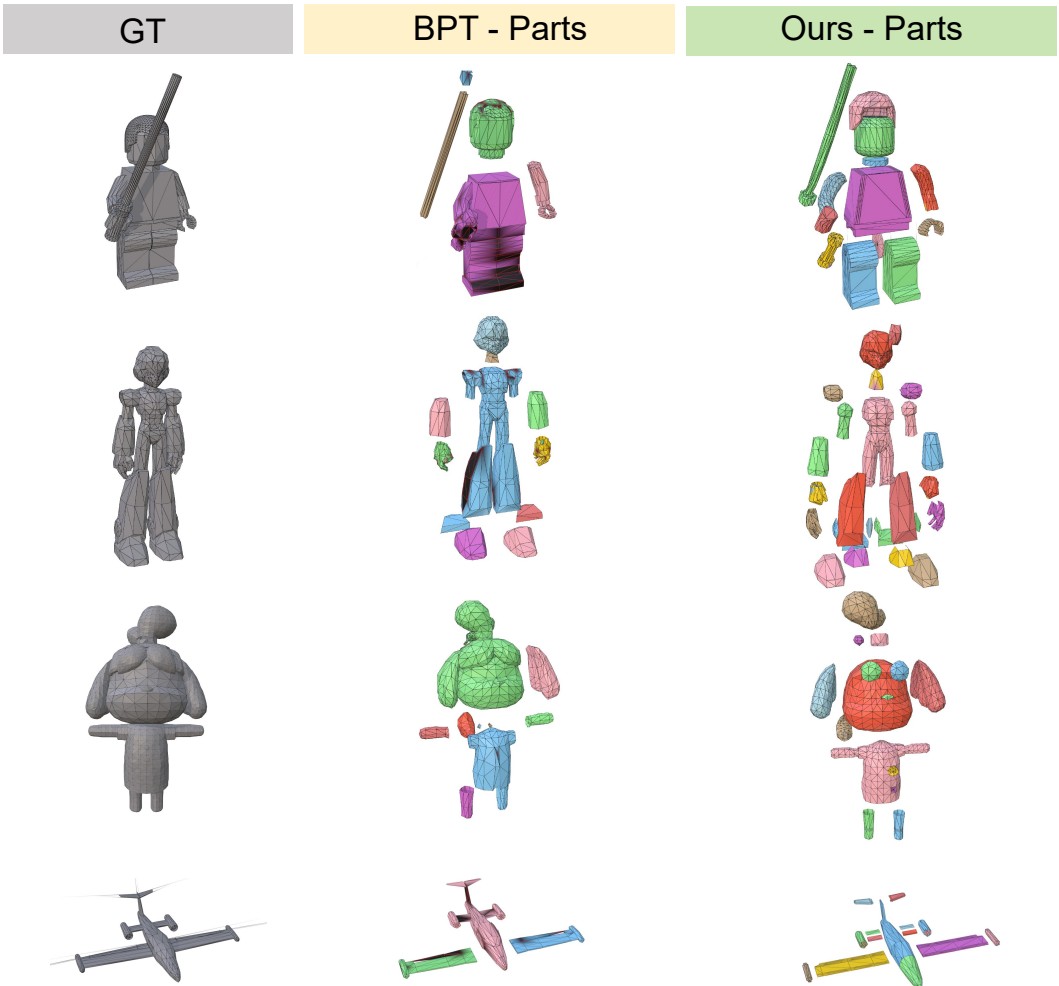

Figure 24: Qualitative comparison with baseline method (BPT Weng et al. (2025)) for object component generation. Different colors denote connected components; non-manifold vertices are colored with black. BPT lacks connected component awareness and generates coupled triangle collections (e.g., indistinguishable LEGO legs and body), while our method natively produces part-aware meshes, substantially reducing manual post-processing efforts for practical applications.

## K  OBJECT PARTS VISUALIZATION

We further visualize the connected components of baseline-generated meshes to better illustrate the superiority of our part-based generation approach. Since baseline methods lack native support for connected component differentiation in generated meshes, we developed a union-find-based algorithm to identify connected components according to vertex connectivity relationships, with color-coding applied for visual distinction.

As illustrated in Fig.24, taking BPT as an example, we demonstrate the advantages of our method in generating mesh components, where connected components are manually displaced for enhanced visual clarity. Notably, owing to the absence of connected component awareness, the majority of connected components in BPT-generated meshes remain coupled (e.g., the legs and torso of the LEGO figure are indistinguishable), necessitating laborious manual post-processing for downstream applications. Conversely, our method natively supports part-aware mesh generation, yielding substantially superior part differentiation compared to BPT.

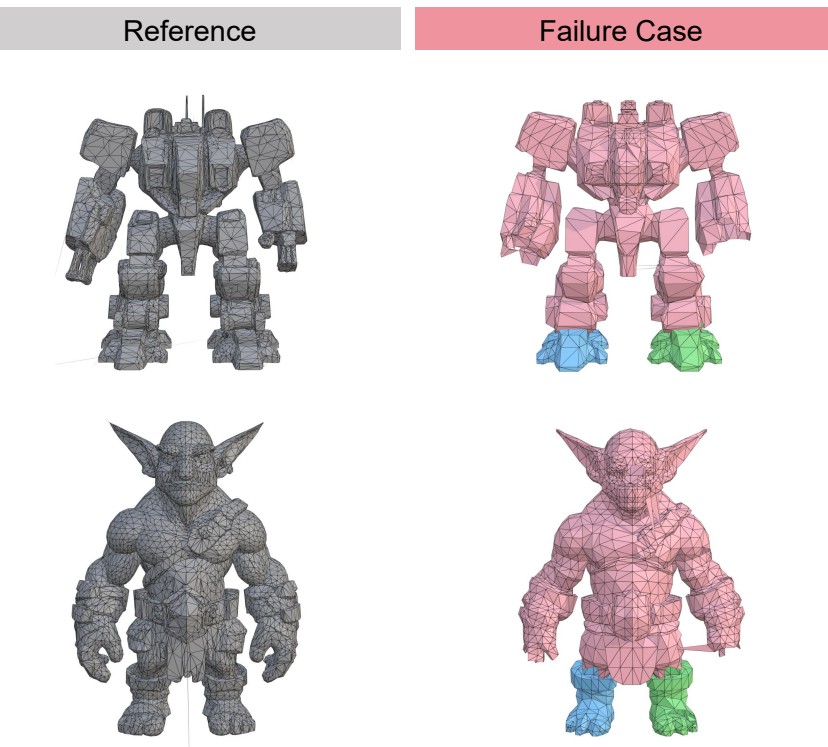

Figure 25: Failure cases illustration.

## L  FAILURE CASE ANALYSIS

We present several failure cases of our current method, as illustrated in Fig.25. Analysis of these cases reveals two main problem of our current implementation:

- To maintain fair comparison with baseline methods, we employ the conventional autoregressive architecture (OPT), which constrains the maximum number of generatable tokens (capped at 10K in our setting). This constraint results in incomplete mesh generation for complex geometries when the token limit is exceeded (e.g., the goblin's face). Adopting more modern architectures such as the Hourglass Transformer (Hao et al., 2024) could substantially alleviate this limitation by enabling significantly higher token capacities.
- For meshes with fine-grained details (e.g., the goblin's belt), our model generates coarser contours relative to reference models. This degradation stems from the 7-bit resolution employed in our current mesh quantization scheme, which fundamentally limits the expressible geometric detail. Training with higher quantization resolution (e.g., 9-bit) is necessary to address this limitation.

## M  LLM USAGE STATEMENT

We used Large Language Models (LLMs) solely for English writing enhancement and proofreading of this paper.

