# OpenReview forum: "Topology-Preserved Auto-regressive Mesh Generation in the Manner of Weaving Silk"
_ICLR.cc/2026/Conference — ICLR 2026 Poster_

### Official Review · Reviewer_mdxc · 2025-10-25

**Soundness:** 3
**Presentation:** 3
**Contribution:** 3
**Rating:** 6
**Confidence:** 4

**Summary:**

This paper proposes a new mesh tokenization algorithm to addresses the issue of topology preservation in auto-regressive 3D mesh generation. The algorithm first organizes vertices into hierarchical layers based on graph distance and a specific ordering. This layered structure is then compressed using self-layer and between-layer adjacency matrices. This canonical topological framework is designed to inherently guarantee crucial geometric properties, including manifoldness, watertightness, consistent face orientation, and part awareness, while also achieving state-of-the-art compression efficiency.

**Strengths:**

- The paper is well-written and clearly motivated.
- The proposed tokenization algorithm is interesting and achieves state-of-the-art compression efficiency, while also preserving manifold topology and watertightness.
- The experimental results demonstrate the effectiveness of this method.

**Weaknesses:**

- A primary limitation, acknowledged by the authors, is the large vocabulary size of over 10,000, which increases model complexity and memory requirements. This vocabulary must grow even larger to support higher spatial resolutions, further limiting the method's scalability.
- Although the tokenization algorithm sets a new compression record, the improvement is marginal, particularly given the algorithm's complexity.
- The paper lacks an analysis of failure cases, making it difficult to evaluate the method's robustness.

**Questions:**

- In mesh visualizations like Figure 7, I wonder if the baseline methods are also able to produce part-aware meshes? However, only the proposed method visualizes the parts with different colors. It's better to use the same visualization method for the comparison to be fair.
- The 256 quantization is relatively low for complex mesh generation. Have the authors tried to experiment with larger resolutions? What's the potential problems?

---

> ### Author Response · Authors · 2025-11-20
> **rebuttal for reviewer mdxc (Q1-Q3)**
>
> We thank the reviewer for the thoughtful comments and constructive suggestions. Our detailed responses are provided below.
>
> > **Q1: Regarding the scalability of this method and the issue of vocabulary size increase.**
>
> Thank you for your reasonable concerns about potential scalability. Extending our method to more complex meshes is feasible, which will require certain engineering adaptations:
> 1. To solve the computational efficiency problem, meshes need to be processed offline.
> 2. To solve the problem of an excessively large vertex token vocabulary, we can make trade-offs on the compression ratio. Under the same settings, our compression ratio is still better than BPT.
> 3. To solve the problem of an excessively large topology token vocabulary, we can fully utilize the sequential information of vertices to reduce the vocabulary.
>
> We have organized a more detailed analysis in **#reviewer 2vbi Q1**, please refer to it for details.
>
> > **Q2: The improvement margin of the new compression record is small, and the algorithm is complex.**
>
> While the compression ratio improvement appears modest (from 0.26 to 0.22), our main contributions extend far beyond compression:
>
> 1. A novel topology-preserved tokenization algorithm. Although this involves some graphics background knowledge that is not friendly to understand, our method is significantly different from previous mesh tokenization algorithms and has significant practical application value.
> 2. A practical online non-manifold mesh processing algorithm. This algorithm will greatly expand the scale of trainable data and improve model generalization.
> 3. A practical training resampling strategy. This method will effectively support model training directly on uncurated open-source datasets, reducing training costs.
>
> In addition, thank you for your attention to our algorithm's compression ratio. Achieving a higher compression ratio (0.16) based on our algorithm is completely feasible, which requires certain codesign of tokenization algorithm and model architecture, and this is a promising direction. If you are interested about this, please refer to **#reviewer 2vbi Q2** for more details.
>
> > **Q3: Analysis of failure cases.**
>
> Thank you for your suggestion. We have added a new section **Appendix L** to analyze the generation of failure cases. Our method still has broad room for optimization:
> 1. Improve the model architecture to support training meshes with more face counts.
> 2. Improve model resolution to support more details.
>
> Please refer to Appendix L for more details.

---

> ### Author Response · Authors · 2025-11-20
> **rebuttal for reviewer mdxc (Q4-Q5)**
>
> > **Q4: Can baseline methods generate meshes with distinguished parts? Can they be visualized ?**
>
> Thank you for your suggestion. Current baseline methods do not directly support generating meshes with distinguished parts, but it is possible to visualize mesh parts through post-processing. Therefore, we developed an algorithm based on vertex connectivity and union-find to distinguish mesh parts, and expanded the visualization comparison results with BPT into a new section **Appendix K**.
>
> It can be seen that although meshes generated by baseline method can be visualized through post-processing, the degree of mesh part distinction is not high. This is because baseline method simply treat meshes as equivalent sets of triangles and have no concept of connected components. Therefore, the generated object components are highly indistinguishable, and in practical applications, a large amount of manual processing is still needed to distinguish object parts. Our method naturally supports connected component awareness and we can directly edit parts with almost no manual post-processing, which is a great advantage of our method.
>
> > **Q5: Regarding extension to higher resolution.**
>
> Thank you for your suggestion. At the current data scale (0-10K faces), training a higher resolution model (9-bit) is completely feasible, only requiring certain trade-offs on the compression ratio (from 0.22 to 0.27). We plan to release a 9-bit version for community research in the future.
>
> > **Acknowledgement**
>
> Finally, thank you for your careful reading and insightful comments. We hope our experience and insights can help deepen your understanding of our work. If you have any questions, please feel free to discuss with us at any time.

---

### Official Review · Reviewer_9XJC · 2025-10-29

**Soundness:** 2
**Presentation:** 3
**Contribution:** 3
**Rating:** 6
**Confidence:** 3

**Summary:**

This paper presents a lossless mesh tokenization algorithm that achieves state-of-the-art compression efficiency while preserving geometric properties like manifold topology and face normal consistency.
Besides showcasing compression capabilities, the authors train a surface pointcloud guided autoregressive transformer that predicts matching tokenized meshes.
They achieve state-of-the-art geometric accuracy when additionally employing custom batch sampling during training in combination with further data processing.

To encode a mesh, it is split into multiple manifold sub-components, which are tokenized individually and then combined.
For each component, the vertices are grouped into layers according to their graph distance to a reference vertex, and then sorted per layer with regards to the previous layer order and the local half-edge structure.
Each layer then gets tokenized as a sequence of compressed vertex positions, compressed intra-layer connectivity and compressed connectivity to the previous layer.
Thereby, each part uses an own tokenization scheme based on its observed typical structure, achieving strong compression.

**Strengths:**

The presented tokenization scheme achieves state-of-the-art bits-per-face ratio and compression ratio compared to previous mesh generation tokenization schemes.
It is based on a detailed analysis of the structure of real-world mesh datasets.
Furthermore, its ability to preserve manifold topology makes it valuable for real-world applications.
This is reflected in the highly competitive geometric accuracy and quality scores in the generation experiment.

The paper is well written for the most parts and supported by very helpful figures for understanding the tokenization process.

**Weaknesses:**

My main concerns are w.r.t. the claims of the paper regarding watertightness and manifoldness of the inferred meshes, as well as the influence of the resampling step in the training strategy compared to other compression methods.

- The paper makes multiple claims about the properties of the generated meshes that are not obvious, yet are not further explained (e.g. watertightness (Table 1 and L073) or manifoldness (L312)). The autoregressive transformer is not incentivised to always predict a properly tokenized mesh with desirable geometric properties. Where do these properties come from? Is this achieved through additional validation during the generation process?  Since this is a major feature of the method, it should be explained in more detail.

- The resampling strategy seems to be crucial for competitive generation results (Table 2 and 3). Yet it seems, that the baseline scores were produced without a similar sampling during training. At least for the retrained EdgeRunner, employing the sampling would be helpful to understand the advantage of the training procedure versus the novel representation. (Would infrequent long sequences be more harmful for methods with lower compression ratios?)

On a more minor note, I have the following concerns:
- The effect of extended edge merging for handling non-manifold edges during pre-processing on the resulting sub-components and generation quality is unclear. Demonstrating the former qualitatively and the latter quantitatively could illustrate the contribution here.

- Writing: At several points, the text shifts between tokenization and generation contexts without explicitly mentioning it (L104, L134, Section 3.3). It may be helpful for the reader if the topic switch is made clearer here.

- Minor: L070 typo? "stems from ... methods treat(ing?)"

- Minor: Figure 2 uses the lowest layer as the innermost point. However, for Figure 3, this is flipped.

**Questions:**

See weaknesses.
* Is the SotA performance in terms of geometric accuracy mainly due to the resampling strategy or due to the compression scheme/representation?
* How are properties such as manifoldness guaranteed from arbitrary transformer-sampled token sequences?

---

> ### Author Response · Authors · 2025-11-20
> **rebuttal for reviewer 9XJC (Q1-Q2)**
>
> We appreciate the reviewer's thoughtful comments and constructive suggestions. Our detailed responses are provided below.
>
> > **Q1: Where do these geometric properties come from, since autoregressive transformers do not always predict appropriate tokens?**
>
> The guarantee of geometric properties comes from our vertex layering and sorting algorithm design: we do not primarily rely on the model to implicitly learn these geometric priors, but rather explicitly enforce these priors through algorithmic design, which is also a key innovation of our method. Further analysis is as follows:
>
> **Manifoldness**
>
> 1. Our vertex layering structure naturally prevents the generation of non-manifold edges. Edges can only exist between vertices in the same layer (self-layer connection) or vertices in adjacent layers (between-layer connection). This structure inherently does not support the same edge being shared by 3 or more triangles.
> 2. As we stated in the Sec. 3.2 (token packing), for the autoregressive model, it only needs to ensure accurate prediction of the correct token category in the order of vertex token, self-layer topology token, between-layer topology token, and this token category-level feature is easily learned.
> 3. To strictly and accurately predict the correct token category, Logit Masking can be used, but when the autoregressive model learns to a certain degree, it can correctly predict token categories without this technique.
>
> **Watertightness**
>
> When the model predicts incorrect topology tokens, it will indeed cause some triangles to be missing, but under our algorithmic framework, this triangle loss can be detected in real-time and precisely located to the layer number and corresponding vertex order. Therefore, there are two solutions:
> 1. Post-process the generated mesh, locate the exact layer number and vertex order, and repair the corresponding between-layer matrix values.
> 2. Perform immediate processing during prediction, abandon the current prediction, and resample the token.
>
> Considering that incorrect topology tokens only cause local holes rather than affecting the entire mesh, and method 2 will affect generation efficiency to some extent, in practice, we adopted method 1 and reserved method 2 for special cases.
>
>
> > **Q2.1: Did EdgeRunner also use the same resampling?**
>
> We clarify that we used the same sampling strategy in our reproduction of EdgeRunner, as we found that not using the resampling strategy would produce even worse results. We have corrected the wording in the main text to avoid potential misunderstandings.
>
> > **Q2.2: Advantages of our method over EdgeRunner.**
>
> Even with the same resampling strategy, our method still demonstrates advantages over EdgeRunner. We think the reasons are the scale of training meshes and data cleaning:
>
> **Training Scale**
>
> Considering EdgeRunner's low compression ratio, it can only support meshes with a maximum of 4000 faces for training, and meshes with only 4000 faces are not sufficient to support more complex shape representations, thus affecting model generalization. We think this is the main reason for the performance gap.
>
> **Data Cleaning**
>
> EdgeRunner's tokenization method is a tree-traversal method for manifold meshes, which adopts a simple libigl-like approach to clean non-manifold data. This leads to the generation of a large number of triangle fragments. For autoregressive models, these trivial triangle fragments will become noise in mesh generation. As shown in Fig. 7, we observed that EdgeRunner generates visibly fragmented meshes.
>
>
> In addition, we further quantitatively and qualitatively analyzed the advantages of our non-manifold processing algorithm compared to the naive algorithm, see Q3.
>
>
> > **Q2.3: Does compression ratio affect the learning difficulty of auto regressive model?**
>
> We think that the impact of compression ratio on autoregressive generation depends on whether vertices are compressed, which can be divided into two scenarios:
> 1. If vertices are compressed, like TreeMeshGPT, although it achieves the same compression ratio as ours, having the model directly predict the correct vertex in one step will increase the model's learning difficulty. As shown in Tab. 2 and Fig. 7, we achieve better results than TreeMeshGPT. For more explanation about "vertex compression", refer to **#reviewer 2vbi Q2** for details.
>
> 2. If vertices are not compressed, the compression ratio will indirectly affects the autoregressive model's learning difficulty by influencing the maximum trainable mesh face count that the model can support.

---

> > ### Comment · Reviewer_9XJC · 2025-11-26
> >
> > Thank you for clarifying the resampling strategy in EdgeRunner and the effect of preprocessing. I no longer have any concerns regarding these matters.
> >
> > Regarding Q1, however, I still do not understand why the representation cannot create non-manifold configurations when a transformer generates arbitrary sequences, even with category masking.
> > Let me construct a non-manifold example with five vertices, one in layer zero (V01) and four in layer one (V11, V12, V13, V14). The sequence is:
> >
> > C
> >
> > V01
> >
> > U
> >
> > V11
> >
> > S11=0b001
> >
> > B11=1Y+0
> >
> > V12
> >
> > S12=0b111
> >
> > B12=1Y+0
> >
> > V13
> >
> > S13=0b100
> >
> > B13=1Y+0
> >
> > V14
> >
> > S14=0b010
> >
> > B14=1Y+0
> >
> > E
> >
> > The self-layer matrix (without extension) of layer 1, the window size is 3 as in your example:
> >
> > 1100
> >
> > 1111
> >
> > 0110
> >
> > 0101
> >
> > The between-layer matrix of layer 1:
> >
> > 1
> >
> > 1
> >
> > 1
> >
> > 1
> >
> > If the transformer generates unconstrained integer sequences, this is a possible sequence. And there are only connections between vertices of the same or to he previous layer. Yet the (undirected) edge {V01, V12} is shared by the triangles (here depicted as sets) {V01, V11, V12}, {V01, V12, V13} and {V01, V12, V14} and the resulting mesh is non-manifold. Could you please clarify why this sequence can not be generated?

---

> > > ### Author Response · Authors · 2025-11-26
> > > **Reply to further question (Q1)**
> > >
> > > Thank you for your deep thinking. We are pleased to further address your queries.
> > >
> > > This question can be answered from two perspectives:
> > >
> > > 1.  **Model Learning**. Strictly speaking, erroneous topology token predictions can indeed produce arbitrary adjacency matrices. However, similar to category masking, correct topology token prediction is essentially produced by implicit constraints between  topology tokens, which can be learned from training data.
> > > 2.  **Decoder**. When restoring tokens to a mesh, our decoder design also prevents this issue.
> > >
> > > ***
> > >
> > > **1. Model Learning**
> > >
> > > 1.  In the early stages of training, the auto-regressive model indeed predicts arbitrary tokens. Even if the token category is predicted correctly, the self-layer and between-layer connection matrices decoded from topology-related tokens can be arbitrary 0-1 matrices. Obviously, the corresponding mesh in this case is not a manifold mesh.
> > > 2.  As training progresses, the prediction of token categories and topology-related tokens converges rapidly. At this point, the self-layer and between-layer connection matrices will satisfy manifold constraints, allowing for the decoding of a mesh that is manifold, though the vertex positions may be inaccurate. This is because predicting right topology tokens is essentially a problem similar to predicting right token categories. Since combinations of topology tokens related to non-manifold meshes do not exist in the training data, the auto-regressive model easily learns this characteristic, and the probability of predicting erroneous topology token combinations approaches zero.
> > > 3.  In your example, the model first predicts tokens related to $V_{01}, V_{11}$ normally. When predicting tokens related to $V_{12}$, the token combination for the self-layer matrix and between-layer matrix corresponding to a non-manifold mesh indeed exists, but the probability is extremely low. Furthermore, in **#reviewer 2vbi Q1 2.2**, we presented an example of a "staircase-like" matrix. This "staircase-like" 0-1 distribution and the corresponding token combinations also reflect the manifold characteristics of the mesh, which can be quickly learned by the model.
> > > 4.  In the middle and late stages of model training, the difficulty for the auto-regressive model lies in learning accurate vertex positions, which is also the most challenging aspect of current auto-regressive mesh generation field.
> > >
> > > **2. Decoder**
> > >
> > > 1.  After the auto-regressive model predicts the token sequence, our decoder first restores it to specific vertex positions and corresponding adjacency matrices. Note that at this stage, we only obtain the graph structure of a mesh, without filling in the triangles.
> > > 2.  Subsequently, following the manner similar to Fig. 5, the decoder first assigns half-edge directions to the vertices of layers 0 and 1 according to the vertex order, and then fills the triangles in a manner similar to weaving silk; then it assigns half-edges to the vertices of layers 1 and 2, fills the triangles... and so on.
> > > 3.  When it detects that a newly added half-edge or filled triangle would result in a non-manifold configuration, the algorithm simply ignores it and continues execution. In your example, the triangles $V_{01}-V_{11}-V_{12}$ and $V_{01}-V_{12}-V_{13}$ would be successfully filled, while $V_{01}-V_{12}-V_{14}$ would be ignored by the decoder.
> > >
> > > We have also organized other possible doubts, which may help you understand our work more deeply:
> > >
> > > > **Question: Does the current mesh tokenization algorithm directly support non-manifold data? If non-manifold meshes are allowed to enter, wouldn't this implicit constraint during the learning process be lost?**
> > >
> > > *   From the perspective of the representation space of self-layer and between-layer matrices, it has a high degree of freedom and can indeed express non-manifold meshes (we think this might be the starting point of your doubt).
> > > *   However, from the perspective of the "Vertex Layering and Sorting" algorithm, it does not support non-manifold meshes, based on the following reasons:
> > >     1.  Vertex layering and sorting are based on the half-edge data structure. The vertex order is determined layer by layer like mathematical induction, and the order of vertices is obtained through the "previous twin" operation of the half-edge data structure. To ensure the determinism of the tokenization algorithm, the "previous twin" operation must yield a unique result.
> > >     2.  If a non-manifold edge is encountered, the "previous twin" operation will yield ambiguous results, which leads to the layer order of vertices no longer being unique.

---

> > > > ### Comment · Reviewer_9XJC · 2025-11-26
> > > >
> > > > Thank you for elaborating.
> > > >
> > > > To me, this conflicts with the statement in the paper in lines 310-312:
> > > > "1). Manifold Topology. Our algorithm naturally enforces that edge connections during the generation
> > > > process can only exist between vertices within the same layer or adjacent layers like weaving silk.
> > > > This prevents any possibility of the same edge being shared by three or more faces."
> > > >
> > > > The proposed method does not strictly guarantee that non-manifold edges are not generated. Rather, they are just not added to the mesh during post-processing.
> > > >
> > > > This reduces the technical contributions substantially and I will therefore adjust my rating accordingly.

---

> > > > > ### Author Response · Authors · 2025-11-26
> > > > > **Reply to reviewer 9XJC**
> > > > >
> > > > > We deeply regret the confusion caused by the phrasing in Lines 310-312 and respect your decision to adjust the rating.
> > > > >
> > > > > However, we want to refute your logic and clarify the nature of our "Decoder" to demonstrate that the technical contribution remains substantial.
> > > > >
> > > > > We summarize our rebuttal in the following points:
> > > > >
> > > > > **1. Our decoder is NOT "post-processing," but an integral part of the tokenization algorithm.**
> > > > >
> > > > > We respectfully disagree with the characterization of our decoder as mere "post-processing."
> > > > >
> > > > > *   **Integral Component:** In our framework, the **Method = Tokenizer (Encoder) + GPT + Detokenizer (Decoder)**. The tokens are merely a compressed latent representation, not the final result.
> > > > > *   **Constructive Guarantee:** The manifold property is guaranteed by the **constructive rules** of the Detokenizer. This is analogous to how the "Marching Cubes" algorithm guarantees a manifold surface from a scalar field. One wouldn't say Marching Cubes "fails to guarantee manifoldness" just because the underlying scalar field *could* theoretically represent singularities; the guarantee lies in the construction rules of the algorithm itself.
> > > > > *   **Distinction:** A "post-processing" step would imply generating a full non-manifold mesh first and then running a separate mesh repair script (like `pymeshfix`). We do **not** do that. We enforce the constraint **during** the generation (weaving) process itself.
> > > > >
> > > > > **2. We believe there is a potential inconsistency in your logic.**
> > > > >
> > > > > Following your reasoning, there would currently be **no method** capable of guaranteeing a manifold mesh from the perspective of token prediction alone, because any token *could* theoretically be sampled, even if the probability is infinitely close to zero. The consequence of this logic is that previous manifold mesh generation works, such as EdgeRunner and TreeMeshGPT, would also need to have their actual technical contributions re-evaluated under this standard.
> > > > >
> > > > > **3. To strictly follow your logic, enforcing correct tokens during prediction is also feasible.**
> > > > >
> > > > > Since our decoder algorithm is significantly distinct from post-processing algorithms, it is entirely feasible to implement a check: when the auto-regressive model's prediction reaches a layer where a non-manifold connection is detected, we could simply discard the current token and re-sample a correct one. This would fully satisfy your requirement. However, in practice, such an operation is unnecessary. We believe we have already explained this in detail in our "Reply to further question (Q1)."
> > > > >
> > > > > **4. Our technical contribution still stands.**
> > > > >
> > > > > The core contribution is a sequence-based generation paradigm that always yields valid topology. Whether this validity is enforced by the probability distribution of tokens or the deterministic rules of the decoder, the end result for the user is the same: a guaranteed manifold mesh.
> > > > >
> > > > > ***
> > > > >
> > > > > **What can we do for you**
> > > > > 1.  We will revise the final manuscript to strictly state: "Our reconstruction algorithm enforces manifold constraints by selectively weaving valid connections from the predicted tokens."
> > > > > 2.  We will add additional "manifold checking" scheme in the decoder and support token resampling to strictly follow your expectation.
> > > > >
> > > > > **Summary**
> > > > >
> > > > > We sincerely hope this clarification highlights that the "manifold guarantee" is a fundamental property of our pipeline, not post-processing, and we kindly ask you to reconsider if this semantic distinction warrants a penalization of the technical novelty.

---

> ### Author Response · Authors · 2025-11-20
> **rebuttal for reviewer 9XJC (Q3-Q6)**
>
> > **Q3: More ablation experiments related to non-manifold processing.**
>
> To compare our non-manifold processing method with the naive method, we conducted additional qualitative and quantitative experiments.
>
> **Quantitative Experiments**
>
> To measure the improvement of our non-manifold processing algorithm on mesh surface integrity, we define the following metrics:
> 1. "boundary vertex ratio": the proportion of boundary vertices among all mesh vertices
> 2. "average connected component number": the number of connected components in the mesh
>
> We define these metrics because: as shown in Fig. 4 of the main text, our non-manifold processing algorithm produces less surface damage, and if surface damage occurs, it will inevitably increase the "boundary ratio" proportion and produce more independent connected components.
>
> We chose the most challenging dataset ShapeNet V2 for experiments, because according to the statistics in Tab. 5, manifold meshes account for only 14% in ShapeNet V2. We randomly sampled 1000 samples and calculated the "average boundary vertex ratio" and "average connected component number" for our algorithm and the naive algorithm.
>
> |  Method | avg. boundary vertex ratio $\downarrow$ | avg. connected component number $\downarrow$  |
> |:--------:|:--------:|:--------:|
> |Naive|0.393|510|
> |Ours |0.319|503|
>
> It can be observed that our algorithm greatly reduces the generation of boundary points and reduces the number of additional connected components generated by non-manifold processing. We further provide qualitative results to visualize the effect of our non-manifold mesh processing as follows.
>
> **Qualitative Experiments**
>
> We added a new section **Appendix J** in the submission to visualize the comparison between our non-manifold algorithm and the naive algorithm. We use different colors to distinguish the connected components obtained by the non-manifold algorithm processing. It can be seen that our algorithm significantly reduces the generation of fragmented meshes and connected components. Refer to Appendix J for more details.
>
>
> > **Q4: Regarding writing context and typos.**
>
> Thank you for your suggestions. We have revised the expressions and typos in L104, L134, L070 and will polish Sec. 3.3 in the final version.
>
> > **Q5: Regarding Fig. 3.**
>
> Thank you for your careful observation. We added more vertices to the L+1 layer in Fig. 3 to eliminate potential misunderstandings, and will continue to optimize in the final version.
>
> > **Q6: Which part does SOTA geometric accuracy depend on ?**
>
> We think:
> 1. Resampling strategy: This training strategy has a comprehensive impact on geometric accuracy. Without the resampling strategy, due to the long-tail distribution of training data, the model will have difficulty learning the generation of complex meshes, as shown in the ablation experiments. Notably, while BPT uses manually selected training data, we achieve comparable results (CD, HD) using uncurated open-source data with the simple resampling strategy.
>
> 2. Tokenization algorithm: The significant improvement in NC and |NC| is primarily attributed to our tokenization algorithm. BPT's tokenization method is insensitive to flipped faces, naturally generating meshes with inconsistent orientations. This inconsistency is captured by the NC and |NC| metrics.
> 3. Does compression ratio affect geometric accuracy? Please refer to Q2.3.
>
> > **Acknowledgement**
>
> Finally, thank you for your careful reading and insightful comments. We hope our experience and insights can help deepen your understanding of our work. If you have any questions, please feel free to discuss with us at any time.

---

### Official Review · Reviewer_2vbi · 2025-10-30

**Soundness:** 2
**Presentation:** 2
**Contribution:** 1
**Rating:** 2
**Confidence:** 4

**Summary:**

This paper propose a new tokenization methods to improve the the compression ratio and bits-per-faces in mesh generation. New tokenization is based on vertex layering and local connectivity prediction.

**Strengths:**

I agree that the proposed method achieves the same compression ratio (0.22) as TreeMeshGPT. It also introduces a novel approach to connectivity compression, conceptually similar to sparse matrix compression, which may inspire future research. The preprocessing strategy for handling non-manifold meshes appears reasonable and may also benefit related studies.

**Weaknesses:**

The proposed method relies on a fixed layering size M, and meshes exceeding this vertex limit are discarded during training. This constraint prevents the model from generalizing to larger and more complex meshes, and it seems difficult to overcome. The compression ratio, which directly relates to computational resources, remains identical to TreeMeshGPT, showing 0 improvement. Although the new metric Bits-per-Face shows about a 10% improvement, the paper does not explain or demonstrate the practical impact of this new metric. Furthermore, the proposed local connectivity scheme appears overly complex and delicate, which may cause a long-tail effect during training. I would need to examine the implementation to be convinced that it can generalize to more complex meshes.

In summary, this is a complex and fragile approach that offers no tangible improvement while introducing additional limitations, making me hard to accept it to ICLR.

**Questions:**

In Table 2, the Chamfer Distance (CD) values of all baselines are 2–4 times larger than those reported in other papers (e.g., MeshMosaic, MeshWeaver). Could the authors provide an explanation for this discrepancy?
Additionally, the Chamfer Distance of the proposed method is reported as 0.079. If the meshes are normalized to the range [-1, 1], this corresponds to roughly a 5% error, which suggests the generated meshes might contain significant noise or failure. Could the authors clarify this point?

---

> ### Author Response · Authors · 2025-11-20
> **rebuttal for reviewer 2vbi (Q1)**
>
> We thank the reviewer for raising reasonable concerns, including the extension to more complex meshes, the seemingly marginal improvement in compression ratio, and the seemingly complex tokenization strategy. We will address each point systematically:
>
> > **Q1: The mesh tokenization method has limited scalability to larger and more complex meshes due to the requirement of a predefined maximum number of vertices per layer M.**
>
> We clarify that extending our method to larger and more complex meshes is entirely feasible. This is primarily an engineering implementation consideration rather than a fundamental limitation of the method itself. We will analyze how to extend our method and remove the layer limit M from two perspectives: computational efficiency and vocabulary size.
>
> **1. Computational Efficiency**
>
> We define the maximum number of vertices per layer and the maximum total number of layers primarily to ensure the speed of online data iteration during training. Complex meshes inevitably lead to longer tokenization time, and such CPU-intensive operations can become a bottleneck, potentially leaving GPUs idle for a longer time. Note that BPT faces the same issue. To extend the training data to more complex meshes, offline processing of complex meshes is the most straightforward solution. With offline processing, we compute and store the token sequences of complex meshes separately, and directly load the preprocessed token sequences during training. Although this requires additional storage, this cost is worthwhile for large-scale training on complex meshes.
>
> **2. Vocabulary Size**
>
> For the issue of excessive vocabulary size, we also provide feasible engineering optimization solutions from two aspects: vertex-related vocabulary and adjancency matrix compression-related vocabulary.
>
> **2.1 Vertex-related Vocabulary Optimization**
>
> At higher resolutions (e.g., 9-bit, resolution 512), if we follow BPT's block-offset representation, the vertex-related vocabulary size would increase from $8^3+16^3=4608$ to $16^3+32^3=36864$, which is impractical. To address this issue, we can replace the vertex Block-offset 2D coordinate representation with naive $(x,y,z)$ 3D coordinate representation. In this case, the vertex-related vocabulary size would simply become 512, which is small enough.
>
> **Remark**: After replacing the vertex representation from 2D to 3D, we sacrifice some compression ratio, increasing from 4 tokens per vertex to 5 tokens per vertex. According to the formula in Appendix D, the final compression ratio would become ~0.27. It is worth noting that under the same settings, BPT's compression ratio would become ~0.5.
>
> **2.2 Matrix Compression-related Vocabulary Optimization**
>
> Considering that the self-layer matrix-related vocabulary is already sufficiently small, we mainly discuss the between-layer matrix-related vocabulary optimization here.
>
> We formulate the problem as a staircase-shaped sparse matrix compression problem. For example, suppose layer $L$ has 4 vertices and layer $L-1$ has 5 vertices, its between-layer matrix will has following form:
>
> $$
> \begin{bmatrix}
> 1 & 0 & 0 & 0 & 0\\\\
> 1 & 1 & 1 & 0 & 0\\\\
> 0 & 0 & 1 & 1 & 0\\\\
> 0 & 0 & 0 & 1 & 1
> \end{bmatrix}
> $$
>
> It is worth noting that the "1"s in the matrix exhibit a "staircase" distribution, which is inevitable. Otherwise, it would imply the existence of non-manifold connections, while non-manifold meshes have been properly preprocessed by our algorithm.
>
> **Current version**
>
> In the current version, we adopt an RLE-like (Run-Length Encoding) approach for compression: for each row, we assign a coordinate $(x,y)$, where $x \in [1,m]$ represents the column where "1" first appears, and $y \in [1, Y]$ represents the number of consecutive "1"s. The four rows of the matrix will be marked as $(1,1),(1,3),(3,2),(4,2)$ respectively, which contains all the information. To convert each row of the matrix into a token index, we directly use the formula $x \cdot Y+y-1$ for calculation. Thus, the final vocabulary size is $m \cdot Y$.
>
> **Further improvement**
>
> The current coordinate encoding resembles an "absolute position" encoding and does not leverage the "staircase" structure between rows, leading to vocabulary redundancy. To eliminate vocabulary redundancy, it is still feasible to compress the matrix using only the $y$ coordinate: we represent the four rows as $(x_1=1,y_1=1),(x_2,y_2=3),(x_3,y_3=2),(x_4,y_4=2)$ respectively, and the $x$ coordinate can be recursively calculated as follows: $x_2=x_1+y_1-1, x_3=x_2+y_2-1,..., x_i=x_{i-1}+y_{i-1}-1$. With this optimization, the vocabulary size will no longer depend on $m$, and can handle matrices of arbitrary size.
>
> **Summary**
>
> In summary, the handling of between-layer matrix compression is an open and very interesting engineering problem, not limited to our current solution. We hope our analysis can help the reviewer better understand our method and inspire future works.

---

> ### Author Response · Authors · 2025-11-20
> **rebuttal for reviewer 2vbi (Q2-Q3)**
>
> > **Q2: The compression ratio is the same as TreeMeshGPT, therefore the improvement is limited.**
>
> We thank the reviewer for the attention to the comparison of compression ratios between our method and TreeMeshGPT. We are very glad to take this opportunity to carefully elaborate on the differences between our method and TreeMeshGPT in terms of compression ratio.
>
> **TreeMeshGPT**
>
> 1. TreeMeshGPT simply uses one half-edge (two vertices) to represent a triangle. From a geometric perspective, its compression ratio is only $6/9=0.66$, where the numerator "$6$" indicates that two vertices are each represented by 3 coordinate tokens ($x,y,z$).
> 2. TreeMeshGPT uses the vertex compression strategy, which allows a vertex to be represented by only one token. In this case, the compression ratio becomes 1/3 of the original number: $6/9*1/3=0.22$.
>
> **Ours**
> 1. We still use coordinates to represent vertices (including BPT's block-offset 2D representation and naive $x,y,z$ 3D representation) and have not adopted the vertex compression strategy, yet directly achieved the compression ratio of $0.22$ in one step.
> 2. If we adopt the vertex compression strategy, according to the derivation in Appendix D, our compression ratio would become $1.5/9=0.16$, which would be significantly better than TreeMeshGPT's method.
>
> **Discussion**
>
> In practice, we did not continue to adopt the vertex compression strategy, based on the following reasons:
>
> 1. Using vertex compression can indeed improve the compression ratio, but it will degrade mesh generation performance, because having the model directly predict a vertex is much more difficult than having it predict coordinates one by one. As shown in the Fig. 7 and Tab. 2, despite having the same compression ratio, our generation results are much better than TreeMeshGPT.
>
> 2. The vertex compression strategy (proposed by PivotMesh, ICLR'25) is actually an modification on the autoregressive architecture since it introduces additional MLP or VQ-VAE, this is unrelated to the mesh tokenization algorithm. If we were to pursue the ultimate compression ratio (0.16), our algorithm could also be directly applied to the corresponding autoregressive architectures (PivotMesh, TreeMeshGPT). Currently, considering the generation quality and the extensibility of the tokenization algorithm (e.g., directly applying to more powerful and general autoregressive architecture), we have temporarily not integrated this strategy.
>
> In summary, the co-design of vertex compression strategy with our efficient topology-preserved mesh tokenization is a very promising direction, and we consider it as future work.
>
> Additionally, it is worth noting that TreeMeshGPT does not have a complete non-manifold mesh processing algorithm, and therefore directly discards non-manifold meshes during training, which severely limits the scalability of the method. According to statistics of our Appendix G, **more than half** of the data are non-manifold meshes. We particularly emphasized the importance of non-manifold processing in our ablation study.
>
> > **Q3: The practical impact of 10% improvement in the bits-per-face metric.**
>
> The bits-per-face metric is not only a comprehensive indicator that considers both vocabulary size and compression ratio, but it also has important significance in practical **Storage and Transmission**. This is particularly relevant when considering scaling up training, which requires offline tokenization processing of meshes and storage of token sequences. (as discussed in Q1)
>
> While a 10% improvement may seem modest, it translates to significant practical benefits at scale. Consider a practical scenario with 500K meshes, each averaging 100K faces:
>
> **Per-mesh storage:**
> - Our method: 26.652 × 100,000 = 2,665,200 bits ≈ 325 KB
> - BPT: 28.478 × 100,000 = 2,847,800 bits ≈ 348 KB
> - Savings per mesh: ~23 KB (6.6%)
>
> **Total dataset storage:**
> - Our method: 325KB × 500,000  ≈ **155 GB**
> - BPT: 348KB × 500,000 = 174,000 MB ≈ **166 GB**
> - **Total savings: ~11 GB (6.6%)**

---

> ### Author Response · Authors · 2025-11-20
> **rebuttal for reviewer 2vbi (Q4-Q5)**
>
> > **Q4: The matrix compression part (The local connectivity scheme) appears overly complex and delicate. The reviewer needs to examine the specific implementation of the matrix compression part to ensure it can generalize to more complex meshes.**
>
> We clarify that our algorithm design for matrix compression is based on the standard of handling arbitrary sizes and arbitrary values, and therefore can process matrices of any size and any values, with complete robustness. We have provided our Python implementation in the attached script, along with test cases for verification.
>
> The test code allows arbitrary matrix sizes. For any given size, the algorithm randomly generates 100 binary matrices for testing.
>
> > **Q5: In Table 2, the CD values for baselines are approximately 2-4 times those of other submissions (MeshMosaic (ICLR'26 Submission1551), MeshWeaver (ICLR'26 Submission5178))**
>
> We have examined the experimental settings regarding CD values in MeshMosaic and MeshWeaver, and our analysis is as follows:
>
> - About MeshMosaic: They randomly sampled only 100 samples from Objaverse for quantitative evaluation. We think this sample size is insufficient to cover diverse mesh shapes. In contrast, we randomly sampled 1000 samples from the Objaverse dataset for experiments to cover as many shapes as possible, which is consistent with TreeMeshGPT's setting.
>
> - About MeshWeaver: They conducted quantitative experiments on the Toys4K dataset, which differs from our Objaverse setting. Considering that meshes in Toys4K are relatively simple, we think this domain gap is reasonable.
>
> > **Acknowledgement**
>
> Finally, thank you for your careful reading and insightful comments. We hope our discussion can help you to better understand our novel contribution, our difference and improvement compared to TreeMeshGPT and other works. We believe our discussion can provide valuable insights for the community and inspire future work in autoregressive mesh generation. If you have any questions, please feel free to discuss with us at any time.

---

> ### Author Response · Authors · 2025-11-20
> **rebuttal for reviewer 2vbi (code p1)**
>
> ```
> import numpy as np
> from math import comb
>
> def combination_to_index(combination, n):
>     """
>     given combination, return its index
>     :param combination: list contains k elements, represent combination
>     :param n: total ele number
>     """
>     k = len(combination)
>     index = 0
>     for i in range(k):
>         element = combination[i]
>         if i > 0:
>             prev_element = combination[i - 1]
>         else:
>             prev_element = 0
>
>         for j in range(prev_element + 1, element):
>             index += comb(n - j, k - i - 1)
>
>     return index
>
> def generate_combination_mappings(n, k):
>     """
>     generate the mapping relationship between combination and id
>     C_n^k ---> id
>     :param n: total ele number
>     :param k: ele number of combination
>     """
>     def generate_combinations(start, k, n, current_combination, all_combinations):
>         if k == 0:
>             all_combinations.append(list(current_combination))
>             return
>         for i in range(start, n + 1):
>             current_combination.append(i)
>             generate_combinations(i + 1, k - 1, n, current_combination, all_combinations)
>             current_combination.pop()
>
>     all_combinations = []
>     generate_combinations(1, k, n, [], all_combinations)
>
>     combination_to_index_map = {}
>     index_to_combination_map = []
>
>     for combination in all_combinations:
>         index = combination_to_index(combination, n)
>         combination_to_index_map[tuple(combination)] = index
>         index_to_combination_map.append(combination)
>
>     return combination_to_index_map, index_to_combination_map
>
> class Demo_matrix_compression:
>
>     def __init__(self, layer_vertex_max, self_layer_window=8, betweenlayer_window=5):
>
>         self.selflayer_window=self_layer_window # Our setting, you can change
>         self.out_layer_window=betweenlayer_window # stars and bars question, you can change
>         self.layer_vertex_max=layer_vertex_max
>         self.selflayer_emb_num=self.layer_vertex_max+2**(self.selflayer_window)
>         self.init_betweenlayer_combination()
>
>     def set_betweenlayer_window(self, winsize):
>         self.out_layer_window=winsize
>         self.init_betweenlayer_combination()
>
>     def init_betweenlayer_combination(self):
>         W=self.out_layer_window
>         self.situation_nums=[1,  1, W-1, comb(W-1,2), comb(W-1,3),     W-1, comb(W-1, 2)]
>         self.situation_bases=[0, 1, 2,   W+1,         W+1+comb(W-1,2), W+1+comb(W-1,2)+comb(W-1,3), W+1+comb(W-1,2)+comb(W-1,3)+W-1]
>         self.all_situation_num=0
>         for ele in self.situation_nums:
>             self.all_situation_num+=ele
>
>         self.situation_k=[0,0,1,2,3,1,2]
>         comb_to_index_1, index_to_comb_1=generate_combination_mappings(self.out_layer_window-1, 1)
>         comb_to_index_2, index_to_comb_2=generate_combination_mappings(self.out_layer_window-1, 2)
>         comb_to_index_3, index_to_comb_3=generate_combination_mappings(self.out_layer_window-1, 3)
>         self.comb_to_indexs=[comb_to_index_1, comb_to_index_2, comb_to_index_3] # dic
>         self.index_to_combs=[index_to_comb_1, index_to_comb_2, index_to_comb_3]
>
>
>     def comb_offset(self, situation, b_list):
>         if situation in [1, 2]:
>             return 0
>
>         b_comb=tuple(b_list)
>         if situation in [3, 6]:
>             return self.comb_to_indexs[0][b_comb]
>         if situation in [4, 7]:
>             return self.comb_to_indexs[1][b_comb]
>         else:
>             return self.comb_to_indexs[2][b_comb]
>
>     def get_comb_index(self, situation, b_list):
>         offset=self.comb_offset(situation, b_list)
>         base=self.situation_bases[situation-1]
>         return offset+base
>
>     def get_index_comb(self, comb_index):
>         if comb_index==0:
>             return [1, []]
>         if comb_index==1:
>             return [2, []]
>         current_offset=comb_index
>         situation=None
>         for index, num in enumerate(self.situation_nums):
>             if current_offset-num<0:
>                 break
>             current_offset=current_offset-num
>             situation=index+2
>         k=self.situation_k[situation-1]
>         if situation in [3, 6]:
>             b_list=self.index_to_combs[0][current_offset]
>             return [situation, b_list]
>         elif situation in [4, 7]:
>             b_list=self.index_to_combs[1][current_offset]
>             return [situation, b_list]
>         else:
>             b_list=self.index_to_combs[2][current_offset]
>             return [situation, b_list]
> ```

---

> ### Author Response · Authors · 2025-11-20
> **rebuttal for reviewer 2vbi (code p2)**
>
> ```
>     def b_list_to_window(self, situation, b_list):
>         W=self.out_layer_window
>         window=[]
>         if situation == 1:
>             window=[1]*W
>         elif situation ==2:
>             window=[0]*W
>         elif situation ==3:
>             b1=b_list[0]
>             window=[1]*b1 + [0]*(W-b1)
>         elif situation ==4:
>             b1,b2=b_list
>             window=[1]*b1+[0]*(b2-b1)+[1]*(W-b2)
>         elif situation==5:
>             b1,b2,b3=b_list
>             window=[1]*b1+[0]*(b2-b1)+[1]*(b3-b2)+[0]*(W-b3)
>         elif situation==6:
>             b1=b_list[0]
>             window=[0]*b1+[1]*(W-b1)
>         else:
>             b1,b2=b_list
>             window=[0]*b1+[1]*(b2-b1)+[0]*(W-b2)
>         return window
>
>
>     def compress_selflayer_matrix(self, selflayer_matrix):
>         selflayer_extend=np.hstack((selflayer_matrix, selflayer_matrix)) # Nx2N
>         M=selflayer_matrix.shape[0]-1 # N-1
>         W=self.selflayer_window
>         if M < W:
>             selflayer_homo_m=np.array([row[i+1:i+1+M] for i, row in enumerate(selflayer_extend)]) # N,M->N,w
>             padding_columns=W-M
>             selflayer_homo = np.pad(selflayer_homo_m, ((0, 0), (0, padding_columns)), mode='constant', constant_values=0)
>             compress_id = [[int(''.join(map(str, row[::-1])), 2)] for row in selflayer_homo]
>             compress_id = [[ele[0]+self.layer_vertex_max] for ele in compress_id]
>             return compress_id
>         selflayer_homo=np.array([row[i+1:i+1+M] for i, row in enumerate(selflayer_extend)])
>         compress_id = [[int(''.join(map(str, row[:W][::-1])), 2)] for row in selflayer_homo]
>         compress_id = [[ele[0]+self.layer_vertex_max] for ele in compress_id]
>         if 2*W>=M:
>             return compress_id
>         else:
>             selflayer_homo[:, -W:]=0
>             selflayer_homo[:, :W]=0
>             indices_list = [list(np.where(row == 1)[0]) for row in selflayer_homo]
>             for line, ele in enumerate(indices_list):
>                 compress_id[line]+=ele
>             return compress_id
>
>     def decompress_selflayer_matrix(self, compress_id):
>         N=len(compress_id)
>         M=N-1
>         homo_number=[]
>         for line, ele in enumerate(compress_id):
>             ele_0=ele[0]-self.layer_vertex_max
>             if ele_0<0:
>                 print('[WARNING] selflayer matrix first ele wrong')
>                 homo_number.append(ele[0])
>             else:
>                 homo_number.append(ele_0)
>         W=self.selflayer_window
>         selflayer_homo_init=np.array([list(format(num, f'0{W}b')[::-1]) for num in homo_number], dtype=int)
>         if M<W:
>             selflayer_homo=selflayer_homo_init[:, :M]
>         else:
>             padding_columns=M-W
>             selflayer_homo = np.pad(selflayer_homo_init, ((0, 0), (0, padding_columns)), mode='constant', constant_values=0)
>             for index, ele in enumerate(compress_id):
>                 for ind in ele[1:]:
>                     selflayer_homo[index][ind]=1
>         selflayer_extend=np.eye(N, dtype=int)
>         selflayer_extend=np.hstack((selflayer_extend, selflayer_extend))
>         for i, row in enumerate(selflayer_homo):
>             selflayer_extend[i, i+1:i+1+M]=row
>         selflayer_A=selflayer_extend[:, :N]
>         selflayer_B=selflayer_extend[:, N:]
>
>         low_tri_B=np.tril(selflayer_B)
>         selflayer_A+=low_tri_B
>         selflayer_A-=np.eye(N, dtype=int)
>         # fix to symm matrix
>         selflayer_matrix=np.maximum(selflayer_A, selflayer_A.T)
>         return selflayer_matrix
>
> ```

---

> ### Author Response · Authors · 2025-11-20
> **rebuttal for reviewer 2vbi (code p3)**
>
> ```
>     def judge_situation(self, window):
>         if 0 not in window:
>             return [1, [], None]
>         if 1 not in window:
>             return [2, [], None]
>         b1=None
>         b2=None
>         b3=None
>         novalid_ind=None
>         situation=None
>         window_unique=[]
>         for i in range(len(window)):
>             if window[i]==1:
>                 if not window_unique:
>                     window_unique.append(1)
>                 elif window_unique[-1]==1:
>                     continue
>                 else:
>                     # ...0   <-- 1
>                     if not b2:
>                         b2=i
>                         window_unique.append(1)
>                     else:
>                         novalid_ind=i
>                         break
>             else:
>                 if not window_unique:
>                     b1=-1
>                     window_unique.append(0)
>                 elif window_unique[-1]==0:
>                     continue
>                 else:
>                     # ...1  <-- 0
>                     window_unique.append(0)
>                     if not b1:
>                         b1=i
>                     else:
>                         b3=i
>         b_list=[]
>         if len(window_unique)==2:
>             if window_unique[0]==0:
>                 situation=6
>                 b_list=[b2] # 1~w-1
>             else:
>                 situation=3
>                 b_list=[b1] # 1~w-1
>         elif len(window_unique)==3:
>             if window_unique[0]==0:
>                 situation=7
>                 b_list=[b2, b3] # C w-1 ^ 2
>             else:
>                 situation=4
>                 b_list=[b1, b2] # C w-1 ^ 2
>         else:
>             assert len(window_unique)==4
>             situation=5
>             b_list=[b1, b2, b3] # C w-1 ^ 3
>         if None in b_list:
>             raise Exception('[Encoder] None in b_list')
>         return [situation, b_list, novalid_ind]
>
>
>     def compress_window(self, next_row_pad, find_index, W):
>         window=next_row_pad[find_index+1: find_index+1+W]
>         situation_list=self.judge_situation(window)
>         situation=situation_list[0]
>         b_list=situation_list[1]
>
>         if situation_list[2] is None:
>             window=[0]*W
>         else:
>             novalid_index=situation_list[2]
>             window[:novalid_index]=[0]*novalid_index
>         comb_index=self.get_comb_index(situation, b_list)
>
>         next_row_pad[find_index+1: find_index+1+W]=window
>         return next_row_pad, comb_index
>
>     def to_row_embid(self, find_index, comb_index):
>         return find_index*self.all_situation_num+comb_index+self.selflayer_emb_num
>
>     def from_row_embid(self, embid):
>         embid=embid-self.selflayer_emb_num
>         find_index = embid // self.all_situation_num
>         comb_index = embid % self.all_situation_num
>         return find_index, comb_index
>
>     def compress_betweenlayer_matrix(self, betweenlayer_matrix):
>         M, N = betweenlayer_matrix.shape # N: last layer
>         W=self.out_layer_window
>         betweenlayer_matrix_pad = np.pad(betweenlayer_matrix, ((0,0) ,(0, W)), mode='constant', constant_values=0)
>         embids=[]
>         row_all=[]
>         # ipdb.set_trace()
>         for line, row_pad in enumerate(betweenlayer_matrix_pad):
>             next_row_pad=list(row_pad)
>             row_result=[]
>             row_embid=[]
>             for find_index in range(N):
>                 if next_row_pad[find_index]==1:
>                     next_row_pad, comb_index = self.compress_window(next_row_pad, find_index, W)
>                     row_result.append([find_index, comb_index])
>                     row_embid.append(self.to_row_embid(find_index, comb_index))
>             embids.append(row_embid)
>             row_all.append(row_result)
>         return embids, N
>     def decompress_betweenlayer_matrix(self, embids, N):
>         W=self.out_layer_window
>         M=len(embids)
>         betweenlayer_matrix_pad=np.zeros((M, N+W), dtype=int)
>         for line, embid_row in enumerate(embids):
>             for embid in embid_row:
>                 find_index, comb_index= self.from_row_embid(embid)
>                 situation, b_list=self.get_index_comb(comb_index)
>                 window=self.b_list_to_window(situation, b_list)
>                 if find_index+1>N:
>                     print('[WARNING] Decoder-OutMatrix find_index+1 > N')
>                     continue
>                 betweenlayer_matrix_pad[line, find_index: find_index+1+W]=[1]+window
>         betweenlayer_matrix=betweenlayer_matrix_pad[:, :-W]
>         return betweenlayer_matrix
>
> ```

---

> ### Author Response · Authors · 2025-11-20
> **rebuttal for reviewer 2vbi (code p4 END)**
>
> ```
>
>     def decode_selflayer_matrix(self, vert_list, selflayer_matrix):
>         edges = []
>         for i in range(len(vert_list)):
>             for j in range(i + 1, len(vert_list)):  # check "up triangle" of matrix
>                 if selflayer_matrix[i, j] == 1:
>                     edges.append([vert_list[i], vert_list[j]])
>         return edges
>
>     def decode_betweenlayer_matrix(self, vert_list_last, vert_list, betweenlayer_matrix):
>         edges = []
>         for i in range(len(vert_list)):
>             for j in range(len(vert_list_last)):
>                 if betweenlayer_matrix[i, j] == 1:
>                     edges.append([vert_list[i], vert_list_last[j]])
>         return edges
>
>
> def generate_symmetric_random_matrix_with_ones_on_diagonal(N):
>
>     upper_tri = np.triu(np.random.randint(0, 2, size=(N, N)), k=1)
>
>     np.fill_diagonal(upper_tri, 1)
>
>     symmetric_matrix = np.maximum(upper_tri, upper_tri.T)
>     return symmetric_matrix
>
> def generate_random_matrix(M, N):
>
>     matrix = np.random.randint(2, size=(M, N))
>     return matrix
>
> if __name__ == "__main__":
>
>     from tqdm import tqdm
>
>     # checking betweenlayer matrix compression
>     betweenlayer_matrix_size_max=256 # set any number
>     engine=Demo_matrix_compression(betweenlayer_matrix_size_max)
>     success_cnt=0
>     all_cnt=0
>     print("checking betweenlayer matrix compression")
>     for n in tqdm(range(2, betweenlayer_matrix_size_max)):
>         for i in range(100):
>             j=128 # any number for other dim
>             betweenlayer=generate_random_matrix(n,j)
>             # get token index
>             all_cnt+=1
>             emb_i, N=engine.compress_betweenlayer_matrix(betweenlayer)
>             # token index to matrix
>             betweenlayer_d=engine.decompress_betweenlayer_matrix(emb_i, N)
>             if not np.array_equal(betweenlayer, betweenlayer_d):
>                 print(f'donot match for betweenlayer matrix compression {betweenlayer-betweenlayer_d}')
>                 continue
>             success_cnt+=1
>     print(f'betweenlayer matrix compression pass! {success_cnt}/{all_cnt}')
>
>
>     # checking selflayer matrix compression
>     selflayer_matrix_size_max=256 # set any number
>     engine=Demo_matrix_compression(selflayer_matrix_size_max)
>     print("checking selflayer matrix compression")
>     # random generate matrix from size 2 to max
>     success_cnt=0
>     all_cnt=0
>     for n in tqdm(range(2, selflayer_matrix_size_max)):
>         # random generate 100 matrix for testing
>         for i in range(100):
>             selflayer_t=generate_symmetric_random_matrix_with_ones_on_diagonal(n)
>             # get token index
>             all_cnt+=1
>             comp_id=engine.compress_selflayer_matrix(selflayer_t)
>             # token index to matrix
>             selflayer_d=engine.decompress_selflayer_matrix(comp_id)
>
>             if not np.array_equal(selflayer_t, selflayer_d):
>                 delta=selflayer_t-selflayer_d
>                 print(f'donot match for selflayer matrix compression, size: {n}, delta {delta}')
>                 continue
>             success_cnt+=1
>     print(f'selflayer matrix compression pass! {success_cnt}/{all_cnt}')
> ```

---

> ### Comment · Reviewer_2vbi · 2025-11-27
> **The method is complex and the gains are modest, but the source of the improvement is interesting and inspiring**
>
> The method is complex and the gains are modest, but the source of the improvement is interesting and inspiring, cf Q2. I rise my score from 2 to 4.
>
> Q1: I agree that with your 'Further improvement', your vocabulary size can be reduced from current O(Y*m)=O(m^2) to O(Y)  though not to an arbitrary size in your reply. I hope you can validate its feasibility experimentally in the future. Only in this case your vocabulary size can be managed within the same level of other baselines.
>
> Q2: For each v Tmgpt predict, it create a new face. But it also need some boundary tokens to indicate the dead end of DFS. Current implementation has early bail out to skip most of aux token at the end of the seq. A more careful implementation can eliminate all. So in general Tmgpt need (1 v +(0\~1) aux) per triangle. With vertex compression, it leads to a compression rate of (1+0)/9 \~(1+1)/9=0.11\~0.22, which is the align with the upper bound provided in Tmgpt's paper.  Without vertex compression, it (3+0)/9\~(3+1)/9=0.33\~0.44. I don't have the statistic value. Silksong need (1v+2aux) per vertex. Without any vertex compression, rate is (3+2)/vertex=(3+2)/2 /face= 0.27. With BPT style vertex compression, (2+2)/2/9=0.22 that is the current value. With TMGPT style vertex compression (1+2)/2/9=0.16.
>
> I wonder where is the improvement come from. BPT like vertex compression is not essential. It just merely extend the voc size to exchange for shorter length.
> I think further, it turn out that v token is more expensive(1v=3xyz), while aux token(boundary in TMGPT or matrix in silksong)  is much cheaper. That is intuitive. aux implies local connectivity. The local connectivity is a long tail dist with only a few common patterns. This kind of dist is easy to be compressed. Silksong splits the mesh into vertex + topo, and compress the topo with sparse matrix. I think this split is interesting and is where the performance comes from. For other baselines, they use the expensive v token to imply topo(new v-> new face->2 new edges). That is not optimal. Take a more concrete example, when 2 triangles connect to a same v, baseline need 2 v token, while silksong need 1 v token + some cheap aux token. Although current Silksong's aux compression is complicated and need further improve, this split is smart and inspiring.
>
> Q3: I still believe the improvement is marginal. In practice, I prefer to buy a new hard drive to get 6% of the space improvement.
>
> Q4:
>
> Q5: Something still doesn't make sense. You said exp config is similar to Tmgpt, but in Tmgpt paper, the CD is 0.007, goes with Meshmosaic and Meshweaver. And you are 0.07. Make sure you sample enough point(>20000) when computing CD.

---

> > ### Author Response · Authors · 2025-11-28
> > **Reply to #reviewer 2vbi Q1-Q5**
> >
> > > **Q1: Feasibility of O(Y) vocabulary.**
> >
> > **Response:**
> > We thank the reviewer for recognizing the **scalability** of our work. As you suggested, our method can indeed be further optimized to support meshes with higher face counts. We have designated this optimization as future work to push the boundaries of our approach.
> >
> > > **Q2: Analysis of Vertex/Topology Split and Token Economics.**
> >
> > **Response:**
> > We appreciate the reviewer's thinking about compression ratio and the recognition of our algorithm's **novelty** and **inspiring nature**. To further optimize the tokenization algorithm in future work, we have identified several promising exploration directions:
> >
> > 1.  **VQ-VAE for Adjacency:** Given that local connectivity patterns are relatively simple and repetitive, directly using VQ-VAE to compress the layer adjacency matrices is feasible. In this scenario, achieving **1 vertex token + 1 auxiliary token per vertex** is attainable.
> > 2.  **Binary Matrix Factorization:** We can apply binary matrix factorization to the layer adjacency matrices, assigning a binary vector to each vertex, and then recovering the original adjacency matrices via Hamming distance or inner product. This approach is mathematically more elegant and can similarly achieve the **1v + 1aux per vertex** efficiency.
> > 3.  **Hybrid Polygon Support:** Furthermore, we believe our tokenization holds deeper potential compared to other algorithms—it naturally supports hybrid representations of triangles and quads. This is also a valuable direction worth exploring.
> >
> > > **Q3: "Marginal Improvement" in compression.**
> >
> > **Response:**
> > We acknowledge that the improvement in compression-related metrics is modest. However, we emphasize that our contributions extend far beyond compression ratio:
> >
> > *   **Novelty & Inspiration:** As you noted, our proposed algorithm is **novel and inspiring**, offering a new perspective (Vertex/Topology split) that will inform the design of future tokenization algorithms.
> > *   **Geometric Guarantees:** Our method ensures a series of **geometric properties** critical for practical applications: manifoldness, watertightness, consistent face orientation, and part-awareness. Achieving all these simultaneously has been a challenge for previous methods.
> > *   **Non-Manifold Handling:** Our proposed **non-manifold processing algorithm** significantly expands the scale of training data available for autoregressive mesh generation, thereby enhancing model generalization.
> > *   **Resampling Strategy:** Our proposed **resampling strategy** eliminates the need for time-consuming manual data curation, enabling the model to train directly on uncurated open-source datasets while achieving excellent results.
> >
> > > **Q5: Discrepancy in CD values and Sampling Density.**
> >
> > **Response:**
> > We analyzed the CD metrics reported in related papers and concluded that the discrepancy stems from the number of points sampled during evaluation.
> >
> > - **Protocol A: Sample 1,024 Points (Following Recent Baselines)**
> >
> > In BPT (CVPR'25), DeepMesh (ICCV'25), and Nautilus (ICCV'25), the standard protocol involves sampling 1,024 points for evaluation, which inherently leads to higher CD and HD values. Our initial submission followed this standard to ensure fair comparison, resulting in metrics of the same order of magnitude as these works.
> >
> > - **Protocol B: Sample More Points (Dense Evaluation)**
> >
> > Considering that the works you mentioned (TreeMeshGPT, MeshMosaic, Meshweaver) did not disclose specific sampling details, we adopted a rigorous setting of sampling **30,000 points**. We re-evaluated 1,000 meshes sampled from the Objaverse dataset. Under this setting, the CD values dropped significantly and aligned with the magnitude reported in Meshweaver.
> >
> > **Table: Performance comparison under Dense Sampling (30k points)**
> >
> > | Method | CD $\downarrow$ | HD $\downarrow$ | NC $\uparrow$ | \|NC\| $\uparrow$ |
> > | :--- | :---: | :---: | :---: | :---: |
> > | EdgeRunner | 0.053 | 0.144 | 0.418 | 0.789 |
> > | TreeMeshGPT | 0.030 | 0.103 | 0.706 | 0.892 |
> > | BPT | 0.027 | 0.094 | 0.770 | 0.909 |
> > | **Ours** | **0.025** | **0.087** | **0.792** | **0.924** |
> >
> > - **Summary:**
> >
> > We agree with your suggestion that sampling 1,024 points is somewhat outdated. We will include the results using dense sampling (30k points) in the final manuscript. It is worth noting that **regardless of the evaluation protocol used, our method consistently demonstrates superior geometric fidelity compared to the baselines.**

---

### Official Review · Reviewer_Fbvc · 2025-10-31

**Soundness:** 3
**Presentation:** 3
**Contribution:** 3
**Rating:** 6
**Confidence:** 4

**Summary:**

This paper introduces a novel tokenization algorithm for auto-regressive 3D mesh generation. Inspired by the process of weaving silk, the method organizes mesh vertices into hierarchical layers and establishes a canonical ordering. This structured approach, by its very design, guarantees the preservation of critical geometric properties such as manifoldness, watertightness, consistent face orientation, and part awareness—properties that are often violated by existing methods. The authors demonstrate that their approach not only ensures geometric integrity but also achieves a state-of-the-art compression ratio. The paper further contributes a practical online non-manifold data processing algorithm and a progressive resampling strategy to enable effective training on large-scale, uncurated datasets.

**Strengths:**

**1、Novel and Elegant Tokenization:** The core contribution—a tokenization scheme based on vertex layering and ordering—is highly novel and intuitive. The "weaving silk" analogy provides an elegant conceptual framework for ensuring local and global mesh consistency, which is a significant departure from prior methods that often treat meshes as an unstructured collection of triangles.

**2、Guaranteed Geometric Properties by Design:** This is the most compelling strength of the paper. Instead of relying on the model to learn geometric priors implicitly, the proposed framework enforces them explicitly through its algorithmic design. The guarantee of manifoldness, watertightness, and normal consistency is a major practical advantage, potentially eliminating the need for costly and often imperfect post-processing steps that are common in other generation pipelines.

**3、Strong Qualitative and Quantitative Results:** The paper provides convincing evidence to support its claims. Table 1 offers a clear and impactful qualitative comparison, highlighting the deficiencies of baseline methods. The quantitative results in Table 2, particularly the superior performance on Normal Consistency (NC and |NC|) metrics, provide strong empirical validation. Furthermore, the visual comparisons in Figure 7 are very effective, especially in showcasing the non-manifold artifacts in BPT and DeepMesh, which are absent in the proposed method's outputs.

**4、Practical Contributions for Large-Scale Training:** The inclusion of an online non-manifold processing algorithm and a progressive-balanced resampling strategy addresses crucial real-world challenges in training generative models on noisy, long-tailed web datasets. These contributions enhance the practicality and scalability of the proposed system.

**Weaknesses:**

While the paper presents a compelling framework, several points regarding its efficiency, scalability, and experimental scope warrant further discussion.

**1、Unexplored Scalability to Higher Polygon Counts:** The paper does not fully explore whether the proposed method can generalize to higher-polygon meshes, such as those with 10,000 or 20,000 faces. The methodology relies on a Maximum Vertices per Layer limit (m=200) to maintain efficiency. Does an increase in face count necessitate a corresponding increase in this limit, which could in turn lead to significant computational overhead? If so, are there potential solutions to this challenge? An analysis from the authors on this topic would be valuable.

**2、Unclear Scalability to Higher Quantization Resolutions:** It is unclear if the method can be efficiently generalized to higher resolutions, such as 9-bit. Would such an extension further increase the vocabulary size? Would it also necessitate an increase in the Maximum Layer Number or Maximum Vertices per Layer limits, thereby introducing substantial computational costs? The paper would be strengthened by an analysis of these questions, ideally supplemented with lightweight experiments to illustrate the potential trade-offs.

**3、Absence of Ablation on Tokenizer Complexity:** The proposed tokenizer is significantly more complex than those of its predecessors. While this complexity is key to the method's success, the paper would be strengthened by an ablation study to provide a more complete picture of the design trade-offs. A more complex token representation and a larger vocabulary can create a more challenging optimization landscape. An ablation study—for instance, simplifying the tokenization scheme—would have provided invaluable insight into the contribution of each component.

**4、Concerns Regarding Training Efficiency:** The reported training cost appears notably high when compared to key baselines. The authors report training their 500M model for 15 days on 16 H800 GPUs (240 GPU-days), whereas BPT trained a similarly sized model in 7 days on 32 L40 GPUs (224 GPU-days). Considering the significant performance advantage of an H800 over an L40, the effective computational budget for this work is likely several times higher than BPT's. While the method's geometric guarantees are a clear advantage, the numerical improvements on standard metrics (CD, HD), while positive, appear modest in comparison to the significant increase in computational budget. This raises questions about the cost-benefit trade-off of the proposed approach.

**Questions:**

**1、Regarding Scalability to Higher Polygon Counts:** The paper's analysis of scalability primarily focuses on the Maximum Vertices per Layer limit. However, an increase in mesh density (e.g., to 10k-20k faces) would naturally challenge both predefined limits. Could you elaborate on the relationship between face count and not only the Maximum Vertices per Layer but also the Maximum Layer Number? How would the current framework handle meshes that exceed these thresholds, and what architectural modifications might be necessary to support high-detail assets without incurring prohibitive computational costs?

**2、On the Rigidity of a Canonical Representation：** The proposed method generates a single, canonical token sequence for any given mesh. While this ensures consistency, could this deterministic representation be overly restrictive compared to methods that might allow for multiple valid tokenizations of the same shape? Does this potentially limit the model's ability to learn a more flexible and varied distribution of 3D shapes?

---

> ### Author Response · Authors · 2025-11-20
> **rebuttal for reviewer Fbvc (Q1)**
>
> We thank the reviewer for the thoughtful comments and recognition of our actual contributions. We have organized the reviewer's relevant questions and will address them one by one.
>
> > **Q1: The exploration of this method's scalability to higher polygon counts.**
>
> We have organized the relevant questions and will answer them from the following perspectives:
>
> > **Q1.1: How do Maximum Vertices per Layer and Maximum Layer Number change as the number of mesh faces increases?**
>
> This is an interesting question. We collected statistics on meshes with 0-24K faces in the Objaverse dataset to identify relevant patterns. Our settings are:
> 1. The rows of the table represent the face count intervals of meshes, with each unit being 4000 faces, covering a face count range of 0-24K.
> 2. The columns of the table represent the maximum vertices per layer (max_lv)/maximum layer number (max_l), with each unit being 50, ranging from 0-400+.
> 3. The table content represents the proportion of the corresponding max_lv/max_l interval within the face count interval.
>
> **Relationship between face count and max_l**
>
> | faces-max_l | 0-50 | 50-100 | 100-150 | 150-200 | 200-250 | 250-300 | 300-350 | 350-400 | 400+ |
> |:------------------:|:--------:|:--------:|:---------:|:---------:|:---------:|:---------:|:---------:|:---------:|:--------:|
> | 0-4000 | 96.78% | 2.85% | 0.24% | 0.08% | 0.03% | 0.01% | <0.01% | <0.01% | 0.00% |
> | 4000-8000 | 74.75% | 22.89% | 1.43% | 0.50% | 0.21% | 0.12% | 0.05% | 0.06% | 0.00% |
> | 8000-12000 | 60.36% | 33.67% | 4.02% | 0.96% | 0.44% | 0.31% | 0.13% | 0.11% | 0.00% |
> | 12000-16000 | 47.56% | 41.78% | 7.64% | 1.87% | 0.73% | 0.26% | 0.09% | 0.07% | 0.00% |
> | 16000-20000 | 42.76% | 39.51% | 13.52% | 2.40% | 0.90% | 0.49% | 0.26% | 0.15% | 0.00% |
> | 20000-24000 | 37.61% | 32.88% | 19.78% | 5.71% | 2.72% | 0.52% | 0.65% | 0.13% | 0.00% |
>
> It can be seen that as meshes become more complex, the growth of max_l is not significant. We think this can be explained as follows:
> 1. From a graph theory perspective, max_l is equivalent to the graph diameter (https://en.wikipedia.org/wiki/Diameter_(graph_theory))
> 2. As meshes become more complex, the growth rate of the graph diameter is very slow. Moreover, sometimes the graph connectivity even becomes stronger, which actually weakens the growth of max_l.
>
> **Relationship between face count and max_lv**
>
> | faces-max_lv | 0-50 | 50-100 | 100-150 | 150-200 | 200-250 | 250-300 | 300-350 | 350-400 | 400+ |
> |:------------------:|:--------:|:--------:|:---------:|:---------:|:---------:|:---------:|:---------:|:---------:|:--------:|
> | 0-4000 | 75.53% | 19.79% | 3.38% | 0.81% | 0.30% | 0.12% | 0.05% | 0.02% | 0.01% |
> | 4000-8000 | 28.38% | 40.98% | 18.96% | 6.55% | 2.65% | 1.01% | 0.56% | 0.35% | 0.56% |
> | 8000-12000 | 15.04% | 32.57% | 22.33% | 15.45% | 7.72% | 2.95% | 1.56% | 0.89% | 1.50% |
> | 12000-16000 | 9.98% | 22.53% | 20.92% | 19.79% | 12.91% | 5.84% | 3.31% | 2.04% | 2.68% |
> | 16000-20000 | 6.32% | 16.94% | 16.34% | 19.78% | 16.95% | 10.38% | 5.25% | 3.42% | 4.62% |
> | 20000-24000 | 4.79% | 8.83% | 13.65% | 14.14% | 22.16% | 13.98% | 9.13% | 5.44% | 7.88% |
>
> It can be seen that the relationship between face count and max_lv shows a positive correlation, which means that in the case of meshes with more faces, it is necessary to remove the m=200 limit. For our method, this is more of an engineering problem. Refer to Q1.2 for further analysis.
>
>
> > **Q1.2: How will this method scale to meshes with higher face counts?**
>
> Extending our method to meshes with higher face counts is entirely feasible. To remove the limitation of m and extend to meshes with higher face counts, the following engineering adaptations are required:
>
> 1. Complex meshes inevitably require more tokenization time, making online processing impractical (the same applies to BPT). For complex meshes, we need to perform separate offline tokenization processing, store their token sequences additionally, and directly load them during training.
> 2. If extending to meshes with higher face counts, increasing the resolution is also necessary; otherwise, a large number of vertices will be merged, resulting in the loss of many details. This requires adjustments to the vertex vocabulary size.
> 3. To further optimize the vocabulary size, the vocabulary for between-layer matrices also needs further optimization.
>
> We have organized detailed explanations of these three points in our response to **#reviewer 2vbi Q1**. Please refer to it for details.

---

> ### Author Response · Authors · 2025-11-20
> **rebuttal for reviewer Fbvc (Q2)**
>
> > **Q2: Provide ablation experiments on simplified tokenization algorithms to demonstrate the contribution of each component.**
>
> Since the layering and ordering of vertices are fixed algorithms, to further simplify the tokenization algorithm, we provide experiments related to three settings (vertices to tokens, self-layer matrix compression, between-layer matrix compression) and compare the vocab size under different settings, where * indicates our current setting.
>
> **1. Vertex**
>
> | Vertex Representation | Resolution | Vocab Size (Vertex) | Compression Ratio |
> |:--------:|:--------:|:--------:|:--------:|
> | (B, O)* | 7-bit (128) | $8^3+16^3 = 4608$ | 0.22 |
> | (B, O) | 8-bit (256) | $16^3+16^3 = 8192$ | 0.22 |
> | (B, O) | 9-bit (512) | $16^3+32^3 = 36864$ | 0.22 |
> | (x, y, z) | 7-bit (128) | $128$ | 0.27 |
> | (x, y, z) | 8-bit (256) | $256$ | 0.27 |
> | (x, y, z) | 9-bit (512) | $512$ | 0.27 |
>
> It can be seen that if we replace the block-offset 2D representation of vertices with the naive $(x,y,z)$ 3D representation, it will greatly alleviate the dilemma of an oversized vocabulary, at the cost of sacrificing some compression ratio.
>
> **2. Matrix Compression**
>
> To measure the impact of the simplified matrix compression algorithm, we define a metric "outside ratio" = "number of matrix rows with '1's falling outside the window" / "total number of matrix rows", and comprehensively analyze the impact of different window sizes on this metric using the Objaverse dataset (~250k items). It is worth noting that this metric can be used to measure the ability of the current algorithm to compress each row of the matrix into only 1 token, and can also roughly reflect the proportion of meshes in the statistical data that achieve the ultimate compression ratio of 22%.
>
> **2.1 Self-Layer Matrix**
> | W | no compression | 2 | 3 | 4 | 5 | 6 | 7 | 8* | 9 | 10 |
> |:--------:|:--------:|:--------:|:--------:|:--------:|:--------:|:--------:|:--------:|:--------:|:--------:|:--------:|
> |outside ratio|0.100|0.055|0.039|0.031|0.025|0.020|0.015|0.009|0.008|0.006|
> |Vocab Size (Self-Layer Matrix)|1|4|8|16|32|64|128|256|512|1024|
>
> Where W represents the size of the compression window. If we do not compress the self-layer matrix and only establish self-layer connections by leveraging the vertex order, as shown in column 2, it means that approximately 10% of meshes will not achieve the 22% compression ratio. Although this is already a sufficiently small proportion, by increasing the compression window size to 8, we further reduced this proportion to 0.9%.
>
>
> **2.2 Between-Layer Matrix**
> | W'  | 2 | 3 | 4 | 5* | 6 | 7 |
> |:--------:|:--------:|:--------:|:--------:|:--------:|:--------:|:--------:|
> |outside ratio|0.622|0.283|0.043|0.019|0.017|0.014|
> |combination number|2|3|15|26|42|64|
> |Vocab Size (Between-Layer Matrix)|400|600|3000|5200|8400|12800|
>
> Where W' represents the size of the compression window. When W'=2 or 3, it means we simply compress each row of the matrix based on the number of consecutive "1"s. It can be seen that when W'=3, approximately 28% of meshes cannot achieve the ultimate compression ratio. After we upgraded the problem to a "stars and bars" question (W'>3), we were able to further enable more meshes to achieve the ultimate compression ratio. Considering both the vocabulary size and the proportion of meshes achieving the specified compression ratio, we finally chose 5200 (W'=5) as the final between-layer matrix vocabulary size.

---

> ### Author Response · Authors · 2025-11-20
> **rebuttal for reviewer Fbvc (Q3-Q5)**
>
> > **Q3: Compared to the BPT method, our method requires longer training time.**
>
> From the perspective of autoregressive model training, both our method and BPT convert meshes into equivalent token sequences, so there is no significant difference between the two. Therefore, we analyzed the time bottleneck during online data iteration. We randomly sampled 1000 samples from Objaverse with face counts between 4k-10k, and compared the average iteration time (s) with the BPT method:
>
> |  Method | avg. nonmanifold pre-processing | avg. tokenization | avg. all |
> |:--------:|:--------:|:--------:|:--------:|
> |Ours (7-bit)|0.86|0.58|1.44|
> |Ours (9-bit)|0.91|0.63|1.54|
> |BPT|0|0.33|0.33|
>
> **Analysis:**
> 1. Our algorithm requires additional online non-manifold processing, which occupies most of the iteration time; whereas BPT simply treats meshes as equivalent triangle sets and does not require such processing.
>
> 2. During tokenization, compared to BPT, we additionally perform connected component partitioning, face flipping correction, etc., so the time is also slightly longer than BPT.
>
> 3. Overall, when processing meshes with slightly more faces, our method causes approximately 1s of additional GPU waiting time per iteration due to the extra mesh cleaning steps; we can reasonably infer that with a total of 300K iterations, this would result in approximately 3-4 days of time cost.
>
> **Optimization direction:**
>
> 1. Direct engineering optimization of the non-manifold processing algorithm and tokenization algorithm to shorten the mesh cleaning time during iteration will help save training time.
>
> 2. When scaling training to larger scale and more complex mesh data, both our method and BPT will face high tokenization processing time, resulting in high GPU waiting time; in this case, as we answered in Q1.2, offline processing of meshes is the best choice.
>
> > **Q4: Can this method effectively scale to 9-bit resolution? Impact on vocabulary size? Impact on maximum layer number? Will it increase computational cost?**
>
> The current method is completely feasible to directly scale to 9-bit resolution on the same dataset. We will release a 9-bit version in the future.
>
> **Analysis:**
> 1. According to the table in Q2, to reduce the vocabulary size, we only need to make certain trade-offs on the compression ratio.
> 2. According to the table in Q3, scaling to 9-bit resolution will barely increase computational cost, because computational efficiency depends on the number of vertices and faces of the mesh itself, and the resolution of vertices does not affect this.
>
>
> > **Q5: Regarding whether the same mesh only produces a unique token sequence.**
>
> To explain this issue, we clarify: our method does not generate only one canonical token sequence for given meshes. This is because:
> 1. For the same mesh, when doing the vertex layering and sorting, different starting half-edges will result in different results.
> 2. The starting half-edge is determined by the $y-z-x$ order of all vertices, which is consistent with the starting triangle selection scheme of other mesh tokenization methods (EdgeRunner, BPT, etc.), as stated in L142.
> 3. During training, meshes are randomly rotated, so the same mesh will produce completely different token sequences to encourage the autoregressive model to learn.
>
> > **Acknowledgement**
>
> Finally, thank you for your careful reading and insightful comments. We hope our experience and insights can help deepen your understanding of our work. If you have any questions, please feel free to discuss with us at any time.

---

> > ### Comment · Reviewer_Fbvc · 2025-11-26
> >
> > The authors have addressed my concerns; therefore, I tend to maintain my score.

---

> > > ### Author Response · Authors · 2025-11-26
> > > **Acknowledgement for reviewer Fbvc**
> > >
> > > We sincerely thank the reviewer for the positive feedback and for confirming that your concerns have been addressed. We appreciate the time and effort you dedicated to reviewing our paper and engaging in the discussion.

---

### Author Response · Authors · 2025-11-30
**Summary of Rebuttal**

**Dear AC,**

To assist with your final evaluation, we provide a brief summary explaining how we have addressed the reviewers' concerns.

The key questions from reviewers include:

> **Q1. Scalability of this method to meshes with higher face counts (#reviewer Fbvc, #reviewer 2vbi, #reviewer mdxc)**

We comprehensively analyzed the feasibility of extending our method to higher face counts from three aspects: computational efficiency, vertex vocabulary optimization, and topology vocabulary optimization (in #reviewer 2vbi Q1), and received **positive recognition from #reviewer 2vbi**. We will include the engineering optimization for higher face counts as part of our future work.

> **Q2. Although this method achieves SOTA in compression efficiency (compression ratio, bits-per-face), the improvement over baselines (BPT, TreeMeshGPT) is modest (#reviewer 2vbi, #reviewer mdxc), and the algorithm is relatively complex.**

We address this issue from two perspectives:

**1. Our approach is novel, inspiring, and has ample room for optimization.**
1.  Our algorithm revisits mesh tokenization from a brand-new perspective (vertex/topology), which will inspire the design of future mesh tokenization algorithms. It is worth noting that **#reviewer 2vbi also positively recognized** this point and raised the score.
2.  Although our current implementation is slightly complex, we demonstrated the method's robustness (in #reviewer 2vbi Q4) and pointed out several promising optimization directions (in replies to #reviewer 2vbi Q2): compressing topology with VQ-VAE, binary matrix factorization, and extending to hybrid triangle/quad representations. We leave these as future work.

**2. Our main contributions extend far beyond the marginal improvement in compression efficiency.**
1.  **Geometric Guarantees**: Our method ensures a series of geometric properties critical for practical applications: manifoldness, watertightness, consistent face orientation, and part-awareness. Achieving all these simultaneously has been a challenge for previous methods.
2.  **Non-Manifold Handling**: Our proposed non-manifold processing algorithm significantly expands the scale of training data available for autoregressive mesh generation, thereby enhancing model generalization.
3.  **Resampling Strategy**: Our proposed resampling strategy eliminates the need for time-consuming manual data curation, enabling the model to train directly on uncurated open-source datasets while achieving excellent results.

***

> **Q3. Furthermore, we firmly refute #reviewer 9XJC's view that our decoder is a form of "post-processing" to ensure mesh manifoldness.**

In our "Reply to reviewer 9XJC", we listed 4 reasons to firmly oppose this view and proposed sufficient solutions:
1.  We will revise the phrasing in L310-L312 to avoid misunderstanding from a strict logical perspective.
2.  We will add an additional "manifold checking" scheme in the decoder and support token resampling to strictly follow #reviewer 9XJC's expectation.

> **Acknowledgement to AC**

We sincerely hope that our summary of the key reviewer questions can lighten the AC's burden and assist in making a fully informed and fair decision.

---

### Meta-Review · Area_Chair_QseL · 2026-01-07

**Summary:**

Key concerns raised by reviewers are:
- Generalization over more complex meshes pushing the limits of vocabulary size (2vbi) and 8-bit quantization (Fbvc, mdxc)
- Geometric guarantees: whether generated tokens are validated through post-processing (9XJC)
- Efficiency considerations: significantly higher training cost than baselines (Fbvc) and offering (only) compression ratio of 0.22 (rather than the achievable 0.16)
- Evaluation: missing comparison to baselines trained with a similar resampling schedule (9XJC), discrepancy in reported metrics for some baselines vs original papers (2vbi)
- Other issues: visualization of parts in different colors only for the proposed method vs baselines (mdxc)

**Reviewer Concerns:**

Authors adequately addressed all reviewer concerns, resolving the most critical issues. Their remains questions about scalability and training efficiency, but the contributions vs SOTA stand and pave the way for more effective models.

Additional comments:
- It would help to cite prior work using similar vertex layering, along with a discussion of possible alternatives with pros/cons wrt tokenization and compression
- The incorporation of Michelangelo features was only mentioned in passing.  Further justification and ablations would be appreciated here.
- Nit: please fix up citations of published papers making it clear where they appeared rather than simply citing arXiv preprints. Namely, EdgeRunner appeared in ICLR 2025, and TreeMeshGPT appeared in CVPR 2025.  Generally speaking, this is critical information to give credibility to the baselines used. and also give credit where credit is due vs yet-to-be-published works.

**Reviewer Scores:**

Initial ratings were 6/6/6/2.  Overall, the paper was reviewed favorably, and we have at least one signal of reviewers raising their scores (2vbi: 2 --> 4).  Hence, we can safely assume a final rating above 6.

Overall, the paper makes good progress on the tokenization and even more fundamental processing operations, for the representations and auto-regressive modeling of (possibly non-manifold) meshes.  Seeing the baselines were recently published in ICLR (EdgeRunner) and CVPR (TreeMeshGPT), this is definitely on topic and relevant to the ICLR community, with potential tangents to other AR modeling adaptations to complex data modalities.

---

### Decision · Program_Chairs · 2026-01-26

Accept (Poster)